# Curiosity shapes spatial exploration and cognitive map formation in humans
Danlu Cen [1] ✉, Eva Teichert[1], Carl J. Hodgetts [1,2] & Matthias J. Gruber [1]

Cognitive maps are thought to arise, at least in part, from our intrinsic curiosity to explore unknown places. However, it remains untested how curiosity shapes aspects of spatial exploration in humans. Combining a virtual reality task with indices of exploration complexity, we found that pre-exploration curiosity states predicted how much individuals *spatially* explored environments, whereas markers of *visual* exploration determined post-exploration feelings of interest. Moreover, individual differences in curiosity traits, particularly Stress Tolerance, modulated the relationship between curiosity and spatial exploration, suggesting the capacity to cope with uncertainty enhances the curiosity-exploration link. Furthermore, both curiosity and spatial exploration predicted how precisely participants could recall spatial-relational details of the environment, as measured by a sketch map task. These results provide new evidence for a link between curiosity and exploratory behaviour, and how curiosity might shape cognitive map formation.

A large literature on hippocampus-dependent navigation has shown that active exploration of novel environments is key to building cognitive maps in non-human animals and humans, which is critical for supporting efficient navigation[1–4]. What is less understood, however, is what cognitive and/or motivational factors drive exploration in the first place—particularly in situations where individuals acquire spatial knowledge in the absence of external reinforcers. Early influential theories suggest that curiosity—the innate desire to seek out novel information—may be one of the primary drivers of exploratory behaviour[5–7], and thus may be central to the construction and updating of cognitive maps[6,7]. Despite this, the impact of curiosity on spatial exploration, and in turn spatial memory, has not been directly tested.

Fundamentally, this link is difficult to address in non-human species, with spatial exploration often seen as a manifestation of curiosity[5,6]. However, a fledgling research field on curiosity in humans now provides new theoretical frameworks for understanding the determinants of curiosity, its neural correlates, and its effect on memory formation[8–12]. This new research field has mostly focused on how curiosity drives the acquisition of semantic (non-spatial) knowledge[13–18]. This knowledge-driven type of curiosity is often referred to as epistemic curiosity[19]. Additionally, some research has explored a more sensory-driven form of curiosity, known as perceptual curiosity, which can be sparked by novel, surprising or puzzling stimuli[19–23]. Both types of curiosity can lead to exploratory behaviours that result in the extraction of semantic knowledge (e.g. refs. 18,24,25, see ref. 26 for review). However, despite these recent findings, the types of spatial exploratory behaviours observed across motile species, as well as how curiosity impacts

spatial memory and cognitive map formation, have received far less attention in human studies[27].

Addressing this gap in the literature, we developed a virtual-world exploration task to investigate the direct relationship between states of curiosity, spatial exploration and the fidelity of cognitive maps (Fig. 1). We hypothesised that high-curiosity states stimulate spatial exploration within novel environments, which in turn lead to more precise cognitive maps of those environments. To examine the effect of state curiosity on spatial exploration, we conducted two experiments in which participants freely explored a series of virtual rooms. These virtual rooms, being novel to all participants, were expected to stimulate varying degrees of curiosity and exploratory behaviours[28]. Before entering the room, participants rated their curiosity (i.e., pre-room curiosity), with the type of to-be-visited room clearly visible in the distance (e.g., museum, library, lounge, etc; see Fig. 1B). Given recent findings showing that the actual interestingness of information (in addition to pre-information curiosity) has a different influence on memory[10,29–32], participants also rated how interested they felt about the room following the exploration phase (i.e., post-room interest). Here, pre-room curiosity and post-room interest capture different aspects of the participants' engagement with the rooms. Pre-room curiosity reflects participants' anticipation and motivation to explore the room before entering, driven by prior information, expectations, or a desire to resolve the uncertainty. In contrast, post-room interest reflects participants' retrospective evaluation of how engaging or intriguing they found the room after exploring it. Interest ratings can be influenced by factors such as the novelty

[1]Cardiff University Brain Research Imaging Centre (CUBRIC), School of Psychology, Cardiff University, Wales, UK. [2]Department of Psychology, Royal Holloway, University of London, London, UK. ✉e-mail: danlu.cen@outlook.com

of objects, their arrangement, and how well they match participants' expectations.

To quantify exploratory behaviour, we utilised roaming entropy (RE), a measure that captures the variability and extent of exploration across different spatial locations (path roaming entropy) and orientations (head-direction roaming entropy) in the virtual environment[2,33,34]. Across both Experiment 1 and 2, we hypothesised that higher pre-room curiosity would predict more extensive spatial exploration, reflected in higher path roaming entropy values, and both pre-room curiosity and post-room interest would predict broader visual scanning behaviours, captured by higher head-direction roaming entropy.

In Experiment 1, we examined the relationship between pre-room curiosity, post-room interest and exploratory behaviours. Building on these findings, Experiment 2 sought to replicate the results of Experiment 1 and expanded our investigation to explore how curiosity and exploration influence cognitive map formation. Critically, we expected that higher states of curiosity and more active exploration would translate into the formation of more accurate and detailed cognitive maps of environments, as revealed through higher fidelity map drawings in the memory test of Experiment 2. Furthermore, the larger sample size in Experiment 2 enabled us to explore the influence of individual differences in trait curiosity on curiosity-based exploration[35].

## Methods
### Participants
Participants were recruited from Cardiff University, with 32 students for Experiment 1 and 60 students for Experiment 2. All participants reported having normal hearing, normal or corrected-to-normal vision, and were unaware of the study's objectives. Participants self-reported their gender as men, women, or non-binary. No participants identified as having undisclosed gender. No data on race/ethnicity were collected. Due to adjustments in the scales measuring curiosity and interest, the first three participants from Experiment 1 were excluded. Furthermore, one participant was removed from analysis in Experiment 1 due to incomplete exploration data, having spent very little time in 12 of the 16 rooms (e.g. only briefly opening the door without fully entering or exploring the room contents). The final sample for Experiment 1 consisted of 28 participants (3 men and 25 women, aged 18–25, mean age = 19.79, SD = 1.70). The sample for Experiment 2 included 60 participants (5 men and 55 women, aged 18–25, mean = 19.6, SD = 1.28). All participants gave written informed consent prior to participation and were compensated with either financial reimbursement or course credits. This research was approved by the ethical committee of the School of Psychology at Cardiff University, Wales, UK.

### Virtual environments and design
We created 18 distinct virtual rooms, with two designated for familiarisation and practice (see Fig. S1 for snapshots of all rooms). All the rooms were uniform in virtual dimensions (16 m × 16 m in virtual space). In order to enhance the rooms' realism and encourage participants' exploration, we deliberately clustered the room with furniture and decorations. Our virtual environments were designed to be comparable in terms of overall complexity and object density. Each room was designed to promote naturalistic exploration, featuring a combination of easily visible layout-defining objects (e.g., sofa, bookshelf) and smaller details (e.g., plate on table, books on lower shelf). To further encourage active exploration, some layout-defining objects were partially obscured (e.g., table behind sofa) or deliberately placed along the same wall as the entrance, preventing full visibility from the initial entry point (see Fig. S2 for detailed layout of the rooms). Crucially, the rooms were designed so they could not be fully mapped from a single viewpoint, requiring participants to move around and explore different areas to build a complete cognitive map[36].

In addition to the virtual rooms, an outdoor setting was incorporated into the design. This setting featured a pier connected to each room by a zigzag pathway. Participants began each trial at the pier, navigating the pathway before entering a room for exploration. While the outdoor environment remained constant throughout the experiment, the rooms varied between trials, and their presentation order was randomised across participants. The virtual environment, including the outdoor settings and rooms, was created or assembled and presented in Unity 3D (version 2019.4.15, Unity Technologies) (see Supplementary Methods for more information about virtual stimuli).

### Apparatus
The experiment was conducted on a desktop PC, with visual stimuli presented on an LCD monitor (1920 × 1080 pixels; 60 Hz refresh rate). Participants used the keyboard to move forward (by pressing "W") or backward (by pressing "S") and the computer mouse to change the angle of the field-of-view and to steer their direction. To enhance the sense of presence and agency, participants heard their footsteps through headphones. Sideward movements were restricted. Movement speed in the virtual world was fixed at approximately 3.4 m/s, to simulate a brisk walking pace and ensure a standardised duration spent on the pathway.

### Procedure
The experiment was structured into three distinct phases: familiarisation, exploration, and a sketch map test for room layout and details (which was conducted only in Experiment 2). At the start of the experiment, participants were informed they would be exploring 16 different rooms. During the familiarisation phase, they explored two example rooms to become acquainted with the procedure. This allowed them to get a sense of the types of rooms and environments they would encounter (e.g. rooms representing real-world settings), as well as the overall scope of the exploration task.

In Experiment 2, participants commenced with the Five-Dimensional Curiosity Revised scale (5DCR)[35] prior to the familiarisation phase. The exploration phase closely followed the familiarisation phase, where participants had to explore all 16 rooms in a randomised order with no specific time limit imposed for each room. In Experiment 2, the sketch map test took place immediately after the exploration phase. Although other memory tests were included in both Experiment 1 and 2 (see Supplementary Methods for details of those tests), these were designed to address independent research questions and will not be reported here.

The trials in the familiarisation and exploration phases followed the same procedure (see Supplementary Methods for more details). Each trial began at the pier, where participants could view the room's label on its front wall (illustrated in Fig. 1B). Upon seeing the label, a question about their level of curiosity towards the room appeared, which participants rated on a Likert scale from 1 ("*Not curious at all*") to 10 ("*Very much curious*"), by inputting their response as an integer number using the keyboard (see Fig. 1B for the interface).

After walking along the zigzag-shaped pathway and reaching the room, participants pressed the "E" key on the keyboard, triggering a 5-s door-opening animation. Once inside, participants were instructed to explore each room freely without time constraints, allowing them to engage with the environment at their own pace. Participants' position in the room (location data) and angle of field-of-view (head direction data) were recorded at a screen frame rate of 60 Hz. When deciding to leave, participants pressed "B", leading to another rating display where they rated how interested they felt about the room on a Likert scale from 1 ("*Not interesting at all*") to 10 ("*Very much interesting*").

Experiment 2 included an immediate memory test, conducted after a brief 5-min break following the exploration phase, to assess participants' recall of spatial details. In this test, participants were asked to sketch the layout of each explored room, with the drawing order randomised across participants to differ from the exploration sequence. Standardised instructions and example sketch maps of the Bridal Shop and Cinema from the familiarisation phase were provided to ensure consistency in map drawing. At the start of each trial, participants saw an image of the room's label (e.g., Bedroom, Lounge, or Classroom), and were given a paper sheet marked with the room's type and an outlined square representing the room's boundaries. They were instructed to include key spatial elements such as

**Fig. 1 | Overview of experimental procedure.**
**A** Experiment flow chart outlines the three sessions in the experiment. The first session, Familiarisation, comprises three trials allowing participants to become accustomed to the movement controls and the general experiment procedure. The second session, Exploration, involves 16 trials divided into four blocks, with short rest intervals permitted between blocks. In the final session, the Memory Test, participants are tasked with drawing the layouts of the rooms that they visited during the Exploration session. Note that the sketch map test only took place in Experiment 2. **B** Sample trial in the Exploration session. This depicts a representative trial from the Exploration session. Participants start at a pier, where they can see the type of room they will visit (e.g., Lounge). Their initial task is to rate their level of curiosity about the room on a Likert scale ranging from 1 ("*not curious at all*") to 10 ("*very much curious*"). They then proceed along a pathway to the room, encountering six objects en route (e.g., kettle and bandshell). Upon entering the room, they are free to explore. When they choose to exit, their final task is to rate the interestingness of the room using a similar Likert scale, from 1 ("*not interesting at all*") to 10 ("*very much interesting*").

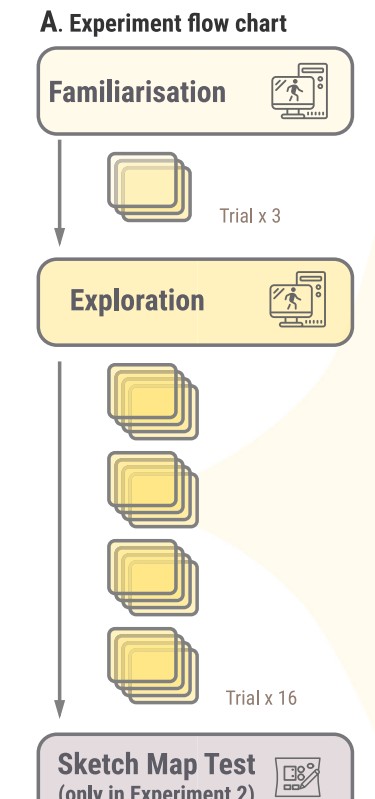

**A. Experiment flow chart**

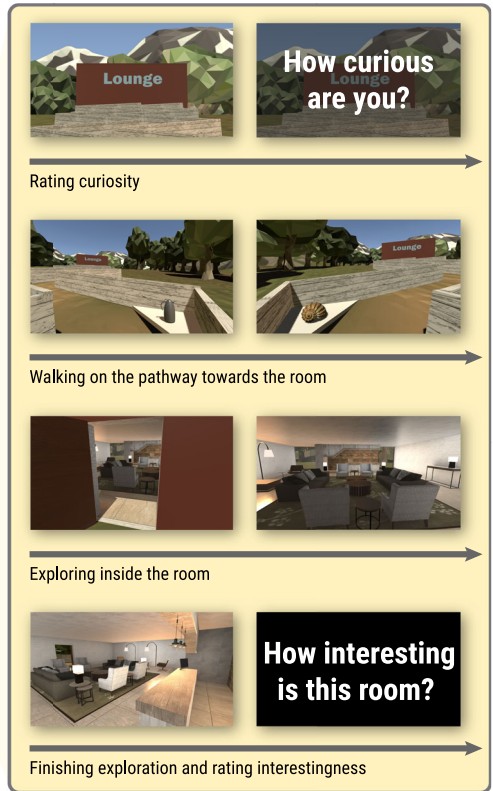

**B. Sample trial in the Exploration session**

---

furniture, doors, and windows in their sketches. Data from 5 participants were excluded from the memory analysis due to minor changes made to the test instructions and the order of the memory test relative to another memory task, implemented after their participation.

### Quantification of exploration

We used roaming entropy to quantify participants' exploration, which has been used in prior animal and human studies to measure the level of exploration[2,33,34]. Roaming entropy is an index of the variability in an individual's territorial coverage, and is computed on how many different locations the participants visit across the entire time of exploration. A high roaming entropy value is achieved when the participant visits and spends equal amounts of time at multiple locations. A low roaming entropy value, in contrast, would be observed when the participant restricted their attendance within a few locations (see Fig. 2C for different variations in roaming entropy). Here, we computed two distinct roaming entropy measures to capture complementary aspects of exploration: *path* roaming entropy and *head-direction* roaming entropy. Path roaming entropy measures spatial exploration by tracking participants' movements through the virtual rooms —specifically, how thoroughly they navigated different locations within each room. A higher path roaming entropy indicates a more extensive coverage, as participants moved through a larger portion of the room. In contrast, head-direction roaming entropy captures visual exploration by recording participants' head direction as they moved within the room. This measure reflects how broadly participants scanned the environment visually, turning their heads to observe their surroundings from different angles (Fig. 2B; and see Fig. 2C for different variations in path roaming entropy and head-direction roaming entropy). Together, these two roaming entropy measures allow us to separately examine how spatial and visual exploratory behaviours contribute to the formation of cognitive maps.

Path roaming entropy was calculated as Shannon's entropy across the available locations in the rooms (32 × 32 grid subtracting those occluded by furniture etc., as illustrated by the pink area in the top panel of Fig. 2B), and head-direction roaming entropy along both horizontal and vertical axes

across 648 directions (18 × 36 grid, as illustrated in the middle and bottom panels of Fig. 2B). We down-sampled the area of each room into a 32 × 32 grid, with each grid cell being 0.5 m × 0.5 m in virtual space. For each room, we labelled those grid cells that were available for attending (i.e., not occluded by furniture etc.) using both artificial intelligence and manually controlled agents in Unity. Next, we projected the trajectory data onto the labelled grid and calculated path roaming entropy as the entropy of the probability distribution of participant $i$'s location falling in a given available grid cell $j$ at a given time $t$:

$$PathRE_{i,t} = -\sum \frac{p_{i,j,t} \log_2 p_{i,j,t}}{\log_2(k)}.$$

In this equation, $k$ is the number of grid cells that were available for visiting. Dividing the entropy by the factor $\log_2(k)$ scales the roaming entropy to the range from zero to unity. Where necessary, the result value was converted to a percentage by multiplying by 100, in order to be comparable with other variables in terms of scale.

Head-direction roaming entropy was calculated in a similar way. We projected the participants' head direction (angle of field-of-view) onto an 18 × 36 grid, horizontally from −180° to 180° and vertically from −90° to 90°. Each grid cell was 10° × 10° in size. We then calculated head-direction roaming entropy as the entropy of the probability distribution of participant $i$'s head facing in a given direction grid cell $j$ at a given time $t$:

$$Head-directionRE_{i,t} = -\sum \frac{p_{i,j,t} \log_2 p_{i,j,t}}{\log_2(k)}.$$

In this equation, $k$ is the number of head-direction grid cells ($k = 648$).

### Scoring of sketch maps

In the sketch map test of Experiment 2, participants sketched maps of the virtual rooms to assess their spatial memory or cognitive maps. Importantly, map drawings allow more insight into the precision and content of memory

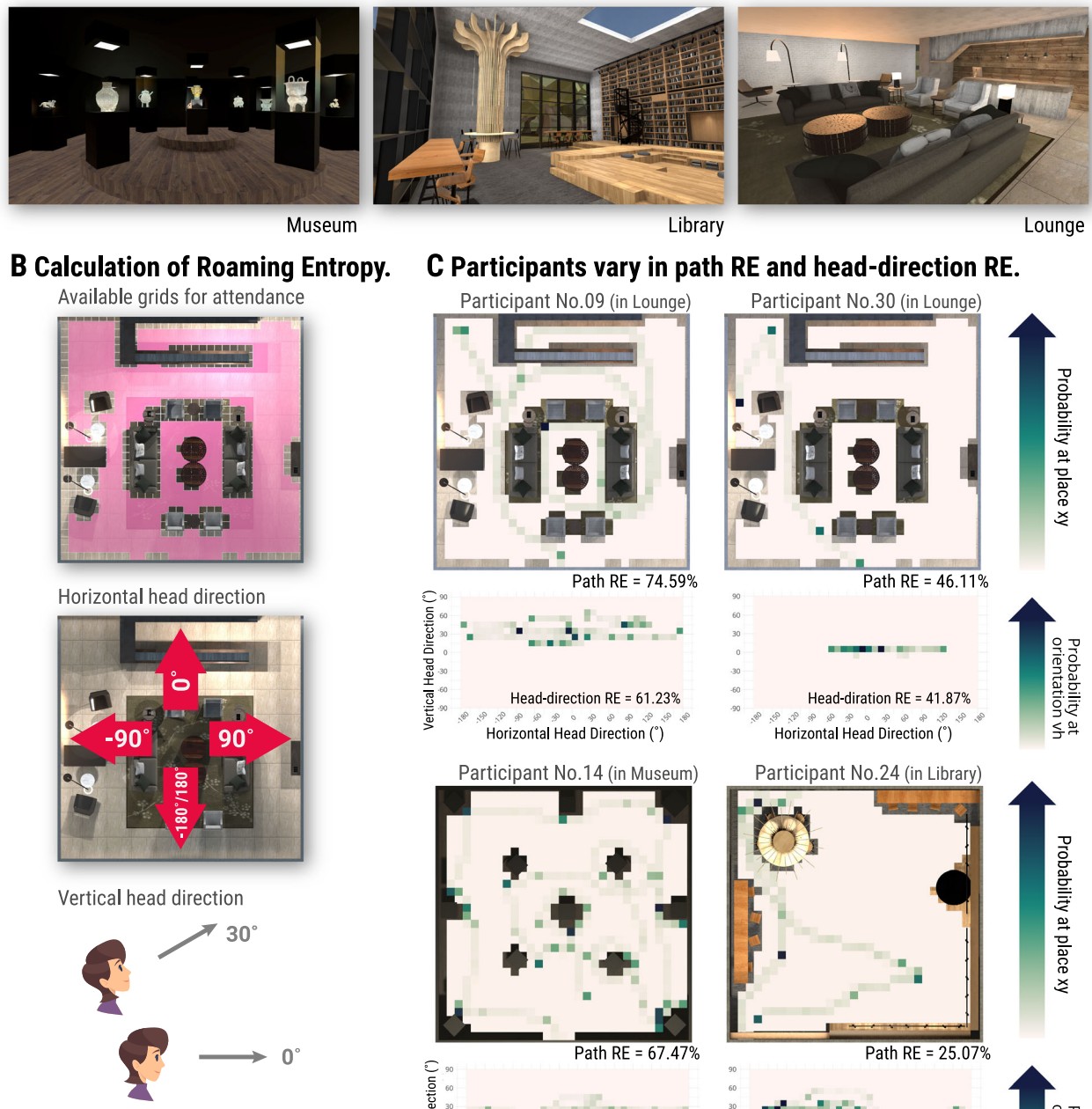

**Fig. 2 | Roaming entropy calculation and sample data. A** Room view snapshots display participant views in three rooms: Museum, Library and Lounge. **B** Exploration trajectory and roaming entropy (RE) calculation. Participant location and head direction data are collected for roaming entropy computation. Path roaming entropy is calculated using a 0.5 m × 0.5 m grid system applied to the room's interior (top panel), focusing on accessible areas (illustrated by the pink overlay in the layout of the Lounge). Path roaming entropy is derived from the probability of participant presence in each accessible grid. Head-direction roaming entropy is computed from head-direction data based on the participant's angle of frame/field-of-view. Its calculation involves two components: horizontal and vertical head direction. The horizontal head direction is relative to the room's entry direction with examples (0°, ±90° and ±180°) depicted in the middle. The vertical head direction is egocentric, where 0° represents the forward-facing direction. Examples of vertical head direction (0° and ±30°) are illustrated in the bottom panel. **C** Location and head-direction probability distribution examples. The upper panels present heat maps of participant location probabilities within the room, alongside corresponding path roaming entropy values. The lower panels show heat maps for head-direction probabilities. Participant data examples: No. 14 in the Museum (high path and head-direction roaming entropy); No. 24 in the Library (low path roaming entropy, high head-direction roaming entropy); No. 09 in the Lounge (high path roaming entropy and head-direction roaming entropy); No. 30 in the Lounge (lower path roaming entropy and head-direction roaming entropy).

representations[37,38]. Based on conceptual frameworks from human neuropsychology[39–41], we developed a scoring system using a sliding scale from 1 (*Poor*) to 5 (*Excellent*), with increments of 0.5. This system was designed to evaluate critical aspects of cognitive map formation, including information about the presence of objects, but critically their spatial properties and relative spatial position. It assessed four distinct dimensions: (i) *Object Presence*, (ii) *Spatial Distortion and Rotation of Features*, (iii) *Relative Positioning*, and (iv) *Spatial Proportion*.

**Object presence.** The dimension focused on the quantity and accuracy of key objects recalled from the virtual room. The focus was on major, layout-defining objects/landmarks (e.g., furniture, doors, windows) that contributed to the overall spatial configuration of the room. While the smaller details (e.g., plates on the table, books on the shelf and cakes in the display case) added natural details to the environment to encourage exploration, they were not considered essential for accurately sketching the room's overall layout. Scores ranged from 1, indicating few or no main objects included, to 5, indicating all or nearly all key objects accurately included. Clustering objects into groups (e.g., drawing a "study area" rather than drawing a desk, chair and filing cabinet separately) was only slightly penalised, as this still demonstrated a grasp of the room's overall spatial layout.

**Spatial distortion and rotation of features.** Here, the size, shape, and orientation of individual objects were assessed. A score of 1 was given for significant distortions or misorientations, while a score of 5 was assigned for high accuracy in these aspects compared to the virtual room.

**Relative positioning.** This criterion evaluated the accuracy of object placement relative to each other. A score of 1 was given when none of the objects or features (including door and windows) were placed correctly relative to others (i.e., randomly placed), while a score of 5 indicated that nearly all objects and features were accurately positioned in relation to each other.

**Spatial proportions.** This dimension assessed the accuracy of distances between objects and features within the sketches. A score of 1 was assigned for completely distorted distances, for instance, if participants depicted room layouts where the distances between objects were highly compressed or if objects were drawn much further away from room boundaries than in the actual virtual room. A score of 5 indicated a highly accurate representation of distances, closely matching those in the virtual room.

Two raters applied this scoring system to evaluate each map across the four dimensions. Each dimension was scored separately, without influencing the evaluation of other dimensions. For example, when evaluating Object Presence, the relative positions of the objects were not considered. Similarly, for the other aspects, evaluations were based solely on the objects presented in the drawing, without penalisation for a lesser number of objects compared to the original room setup. More importantly, the maps were scored based on predefined criteria for accuracy, focusing on spatial relationships and room layout, such as the correct identification and placement of key objects and the spatial relationships between them. This approach minimised the influence of drawing ability and emphasised the cognitive representation of the spatial environment, as the goal was not to critique artistic skill but to evaluate the accuracy of the participant's spatial memory of the virtual room.

Additionally, the two raters performed their evaluations independently of each other, with periodic calibration sessions to discuss ratings, resolve discrepancies, and align their scoring approaches. If inter-rater reliability for certain rooms was low, those maps were re-evaluated and re-coded. To streamline the process, we developed a bespoke online coding platform (https://map-scoring.vercel.app/; see https://osf.io/s2ja7/), which facilitated efficient coding, managed scoring records, and supported calibration sessions by enabling transparent and organised scoring.

Building on previous measures for sketch map drawings[42–46], our coding scheme to the participants' drawings (see Supplementary Methods) allowed us to assess how curiosity states enhance cognitive map formation of novel rooms.

## Statistical analysis

We utilised Bayesian multilevel models (and multivariate multilevel models when necessary) to account for participant-level variability and potential correlations between outcome variables. These models allowed us to include random effects, capturing individual differences, and handle correlations between outcomes. Bayesian models are well-suited for complex, hierarchical data structures, effectively addressing uncertainty and interactions at different levels[47,48]. In our study, the data were nested within participants, and the Bayesian multivariate multilevel approach allowed us to account for these nested relationships and their interactions[47,49]. In addition, data distribution was assumed to be normal, but this was not formally tested.

In our multilevel modelling approach, we centred predictors around the individual mean to specifically examine relationships at the intra-individual level, which aligns with our research focus. This decision was based on both theoretical considerations and empirical evaluation of model fit[50]. For instance, centring predictors on pre-room curiosity, post-room interest, and spatial exploration behaviours (in the model on cognitive map formation) allowed us to isolate the within-person dynamics of these variables, enhancing the model's sensitivity to individual fluctuations. This approach was critical for addressing our key research question—how pre-room curiosity and post-room interest relate to exploration and memory formation within individuals. To further disentangle the effects at both intra- and inter-individual levels, we also included average values for the predictors centred around the grand mean. This separation enables us to account for inter-individual differences while maintaining our primary focus on intra-individual relationships.

We used the 'brms' package[51,52] in R (R Core Team, 2023) to specify and fit our models. Each model was tailored to test specific hypotheses related to exploration measures or the memory test performances. For instance, we modelled the composite score of cognitive map formation as a function of participant-specific factors such as pre-room curiosity levels and post-room interest levels.

Bayesian estimation methods were employed for parameter inference. We ran multiple chains ($N = 4$) with a sufficient number of iterations ($N = 4800$ per chain) to ensure convergence, using standard diagnostics like trace plots and Gelman-Rubin statistics. We selected conservative, weakly informative priors for the parameters in our models to regularise the estimates. Specifically, we used normal priors centred at 0 for the main effects and interactions, reflecting scepticism about large effects. This choice helps to stabilise the estimates and ensure proper convergence during model fitting[47,48].

Posterior distributions of the model parameters were examined to interpret the results. We used posterior means and 93% Highest Posterior Density Intervals (HPDIs) to summarise the effects of predictors. The choice of 93% HPDI is meant to provide a summary of the distribution rather than a threshold for hypothesis testing. To present a comprehensive view of the data, we plotted the entire posterior distributions for parameters of interest, illustrating the relative plausibilities of each parameter value. To ensure the robustness of our models, we conducted posterior predictive checks and sensitivity analyses. These procedures included comparing the models' predictions to the observed data and assessing the influence of alternate priors and model specifications on the outcomes and the overall model fit.

This study was not preregistered.

## Reporting summary

Further information on research design is available in the Nature Portfolio Reporting Summary linked to this article.

## Results

Our virtual rooms successfully elicited varying levels of pre-room curiosity and post-room interest ratings (see Figs. S3 and S4 for the distribution of ratings within and across participants, respectively), as well as path roaming entropy and head-direction roaming entropy (see Fig. S5 for distribution of path roaming entropy and head-direction roaming entropy across

participants). Additionally, we observed a positive correlation between path roaming entropy and head-direction roaming entropy, both across and within participants, indicating that those who explored the space more extensively also tended to visually scan in more detail (see Supplementary Note 1 for further details on the relationship between path roaming entropy and head-direction roaming entropy). Similarly, curiosity and interest ratings were positively correlated, suggesting that participants who reported higher levels of curiosity also found the rooms more interesting (see Supplementary Note 2 for details).

## States of curiosity and interest affect different aspects of exploration

We first examined how states of pre-room curiosity and post-room interest shape exploration. Based on the differential effects of states of curiosity and interest on learning, we predicted that trial-by-trial variation in states of pre-room curiosity and post-room interest would be associated with different aspects of exploratory behaviour. That is, states of pre-room curiosity are expected to affect the magnitude of spatial exploration (i.e., higher pre-room curiosity states are associated with higher path roaming entropy values than rooms with lower pre-room curiosity states). In contrast, rooms that elicit states of higher interest (i.e., participants report that the rooms were more interesting than others) are expected to also be more visually explored than rooms with lower post-room interest states (i.e., higher head-direction roaming entropy, as well as higher path roaming entropy).

We fitted a Bayesian bivariate multilevel model, focusing on path roaming entropy and head-direction roaming entropy as the two outcomes. This bivariate approach was chosen to accommodate the correlation between these two exploratory behaviours after accounting for the predictors in the model (Experiment 1: posterior mean of residual correlation estimates = 0.18, 93%-HPDI = [0.10, 0.27]; Experiment 2: posterior mean of residual correlation estimates = 0.34, 93%-HPDI = [0.28, 0.39]). Note that 93% HPDI provides a summary of posterior distribution rather than a threshold for hypothesis testing. Here, curiosity and interest ratings were introduced as the predictors. Both average ratings (between participants) and trial-level ratings (within participants) were modelled to capture relationships between ratings and roaming entropy at the inter- and intra-individual levels, with trial-by-trial effects nested within participants. Additionally, we controlled for the duration spent within each room for its potential confounding effects ($M$ = 30.35 s, SD = 19.19 s) (see Fig. S6 for distribution of durations spent in each room).

In Experiment 1 ($N$ = 28), the model revealed differentiable effects of curiosity and interest on path roaming entropy and head-direction roaming entropy respectively at the intra-individual level (Fig. 3A, top row). Specifically, path roaming entropy exhibited a positive association with pre-room curiosity (posterior mean of predictor weight $\beta_{curiosity}$ = 0.0059, 93%-HPDI = [0.0014, 0.0104]), and a tentative negative association with post-room interest (posterior mean of $\beta_{interest}$ = −0.0040, 93%-HPDI = [−0.0089, 0.0012]). In contrast, head-direction roaming entropy demonstrated a positive association with post-room interest (posterior mean of $\beta_{interest}$ = 0.0048, 93%-HPDI = [0.0016, 0.0080]), and an indeterminate relationship with pre-room curiosity (posterior mean of $\beta_{curiosity}$ = −0.0005, 93%-HPDI = [−0.0030, 0.0020]) (marginal posterior distributions of these two parameters are visualised in Fig. 3B, top row). Furthermore, the association with pre-room curiosity and post-room interest was different between path roaming entropy and head-direction roaming entropy (posterior mean of difference in pre-room curiosity = 0.0064, 93%-HPDI = [0.0017, 0.0110]; posterior mean of difference in post-room interest = −0.0088, 93%-HPDI = [−0.014, −0.0030]).

## Replication of double dissociation between curiosity-path roaming entropy and interest-head direction roaming entropy in a larger sample

These findings were corroborated in Experiment 2 using a larger sample ($N$ = 60) and identical Bayesian priors for the analysis. Again, duration

within each room was included as a covariate ($M$ = 33.10 s, SD = 22.82 s) (see Fig. S6 for distribution of durations spent in each room). Consistent with Experiment 1, path roaming entropy again was positively associated with pre-room curiosity (posterior mean of $\beta_{curiosity}$ = 0.0039, 93%-HPDI = [0.0003, 0.0073]), and a more tentative negative relationship with post-room interest (posterior mean of $\beta_{interest}$ = −0.0023, 93%-HPDI = [−0.0062, 0.0016]). Head-direction roaming entropy again demonstrated a positive association with post-room interest (posterior mean of $\beta_{interest}$ = 0.0058, 93%-HPDI = [0.0034, 0.0082]), and an indeterminate relationship with pre-room curiosity (posterior mean of $\beta_{curiosity}$ = −0.0009, 93%-HPDI = [−0.0028, 0.0011]) (see Fig. 3A, bottom row). Note, that if we control for room type in our model, the relationship between post-room interest ratings and head-direction roaming entropy remained across both experiments (see Supplementary Note 3 and Fig. S7). The association with curiosity and interest was also different between path roaming entropy and head-direction roaming entropy (posterior mean of difference in pre-room curiosity = 0.0048, 93%-HPDI = [0.0013, 0.0083]; posterior mean of difference in post-room interest = −0.0081, 93%-HPDI = [−0.0120, −0.0041]). Thus, across the two experiments, our findings suggest a double dissociation between the effects of pre-room curiosity and post-room interest on different types of explorative behaviour.

## Stress tolerance boosts curiosity-driven exploration

Psychological theories suggest that there is a cognitive equivalence of curiosity states and traits[53,54], implying that individuals high in trait curiosity experience and express more often states of curiosity. Therefore, we next explored whether the observed positive relationship between state curiosity and spatial exploration could be driven by individual differences in trait curiosity. In Experiment 2, participants were administered the Five-Dimensional Curiosity Revised (5DCR) scale[35] and we focused on the effects of the four subscales that are related to curiosity-induced exploration of novel environments – Joyous Exploration, Deprivation Sensitivity, Stress Tolerance and Thrill Seeking. These subscales' scores, along with their interaction with pre-room curiosity, were incorporated into our Bayesian multilevel model. Among the subscales, *Stress Tolerance* strengthened the positive link between pre-room curiosity and path roaming entropy (posterior mean of interaction weight = 0.0041, 93%-HDPI = [0.0012, 0.0068]) (see Fig. 4). Additionally, *Deprivation Sensitivity* appeared to have a potentially positive effect on this relationship (posterior mean of interaction weight = 0.0025, 93%-HPDI = [−0.0003, 0.0054]) (see Fig. 4A). The findings suggest that the positive correlation between pre-room curiosity and path roaming entropy is moderated by individual differences in trait curiosity. Specifically, individuals with a greater self-reported ability to cope with the anxiety/uncertainty of new situations (*Stress Tolerance*) (Fig. 4A, B) exhibit a stronger relationship between pre-room curiosity and the extent of their spatial exploration.

## Both spatial exploration and state curiosity enhance cognitive map formation

Finally, in Experiment 2, participants were asked to draw sketch maps of the rooms following the exploration phase. This provided a rich source of memory data to investigate the effects of curiosity and exploration on cognitive map formation. We developed a scoring system that allowed us to evaluate different mnemonic aspects that are thought to be critical for cognitive map formation: Object Presence (OP), Spatial Distortion and Rotation of Features (SD), Relative Positioning (RP) and Spatial Proportion (SP) (see Methods). Two raters scored each domain independently, and the inter-rater reliability coefficients (based on two independent raters) ranged from 0.70 to 0.78. Our Bayesian analysis revealed a Cronbach's Alpha of 0.93 (posterior mean) with a 93% HPDI of [0.92, 0.94], indicating a high level of internal consistency for the four aspects. To generate a composite score reflecting the precision of a cognitive map for each room, we averaged the ratings across the four dimensions. A high composite score indicates a high-fidelity cognitive map that accurately captures the presence of layout-defining objects and their spatial relationships.

## A. Differential relationship between curiosity/interest and path/head-direction REs

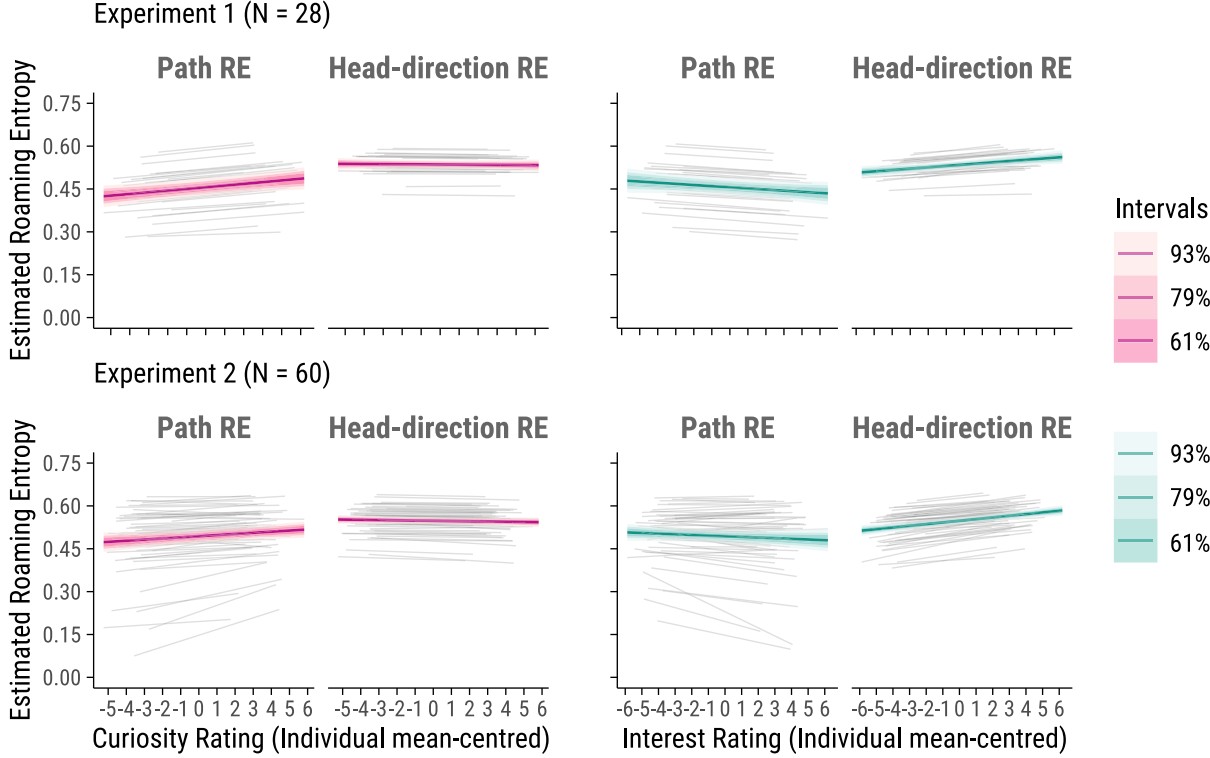

## B. Posterior distributions of weights of ratings

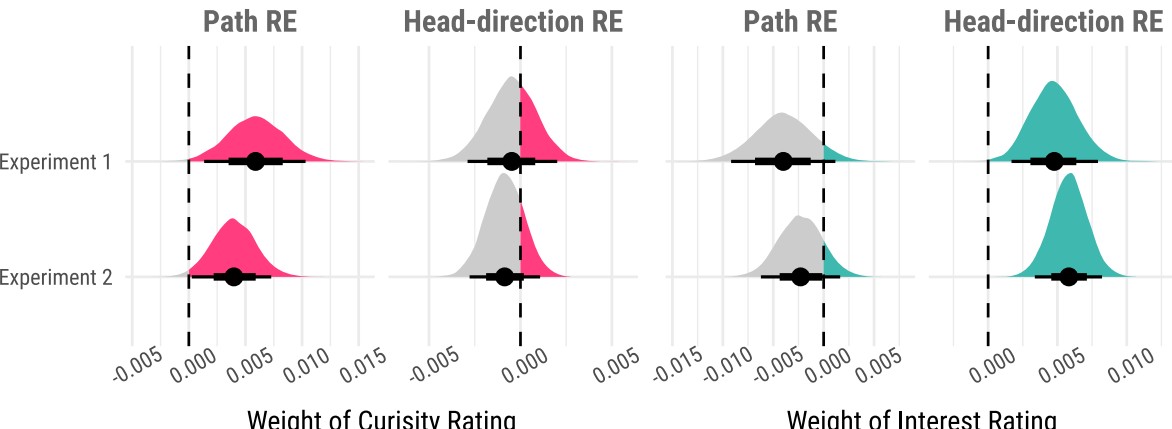

**Fig. 3 | Differential effects of pre-room curiosity and post-room interest on roaming entropy. A** Estimated relationship between curiosity/interest ratings and roaming entropy (RE), with path roaming entropy (left panels) and head-direction roaming entropy (right panels) as outcomes. The relationship is shown separately for curiosity (left column) and interest (right column) in both Experiment 1 (top row) and Experiment 2 (bottom row). Ratings are adjusted to each individual participant's mean. For group-level trends, thick coloured lines represent the average relationship across the participant pool, highlighting the general pattern observed in the experiment. Accompanying these lines, shaded areas show Highest Posterior Density Intervals (HPDIs) at 93%, 79% and 61%, representing descending levels of certainty. Individual-level trends are shown using thin grey lines, demonstrating the variability in the relationship between ratings and roaming entropy across different participants. These lines help visualise the extent to which individual responses vary from the average trend, offering insights into the diversity of participants' exploration behaviour in response to their curiosity and interest. **B** Posterior distributions of the weights for curiosity and interest ratings. The influence of curiosity (left column) and interest (right column) ratings on path roaming entropy (left panels) and head-direction roaming entropy (right panels) is illustrated through posterior distributions. These distributions are depicted by shaded shapes, where the width of each shape indicates the level of uncertainty in the weight estimate. Black dots mark the posterior mean weights, and horizontal bars indicate 67% (thick) and 93% (thin) HPDIs. Dashed vertical lines at zero serve as a reference for no relationship.

## A. Impact of trait curiosity on state curiosity-path RE relationship
### Posterior Distributions of Interaction Weights

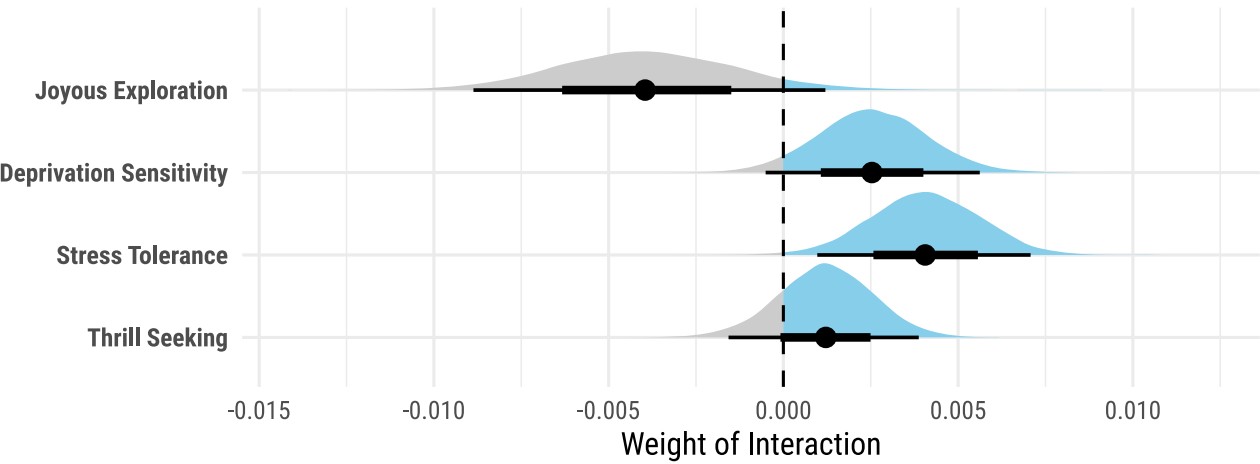

## B. Illustration of impact of stress tolerance on state curiosity-path RE relationship

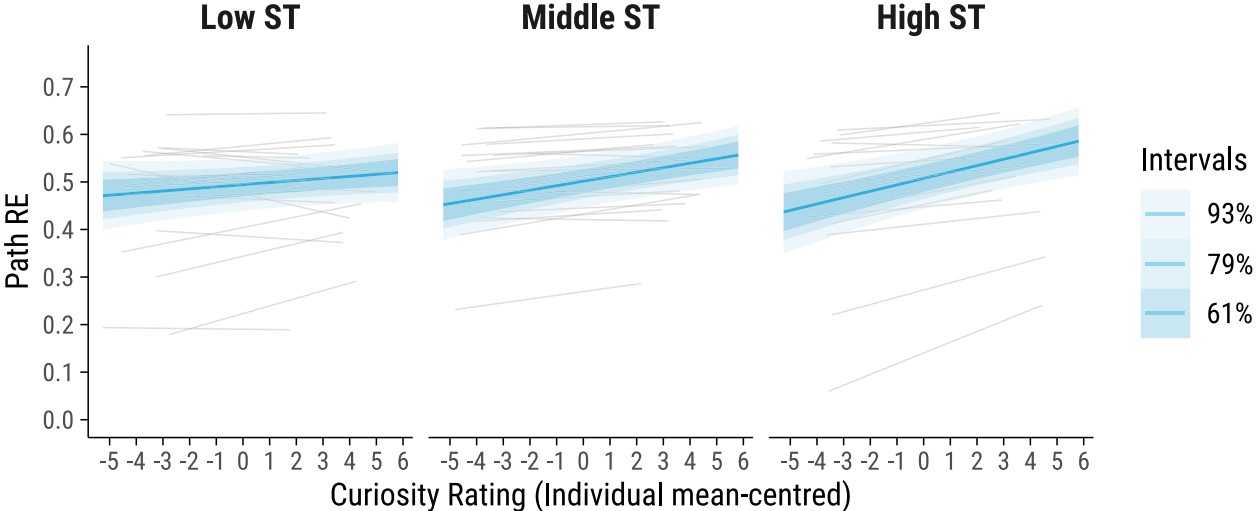

**Fig. 4 | Impact of trait curiosity on the relationship between curiosity rating and path roaming entropy. A** Interaction weights for trait curiosity dimensions. Posterior distributions for the interaction between trait curiosity scores—Joyous Exploration, Deprivation Sensitivity, Stress Tolerance, and Thrill Seeking – and state curiosity ratings on path roaming entropy are illustrated through shaded shapes. The spread of the shapes indicates the level of uncertainty in the interaction weight estimate. Black dots signify the posterior mean weights, while horizontal bars indicate 67% (thick) and 93% (thin) HPDIs. Dashed vertical lines at zero provide a baseline for no interaction. **B** State curiosity and path roaming entropy relationship by Stress Tolerance. Estimated relationship between curiosity ratings and path roaming entropy are stratified by Stress Tolerance (ST) levels: Low ST (scores <= 3.5, $N = 21$), Middle ST (scores between 3.5 and 4.5, $N = 21$), and High ST (score > 4.5, $N = 18$). Curiosity ratings are mean-centred for each individual participant. Group-level trends are shown by thick coloured lines, with 93%, 79% and 61% HPDI shaded areas signalling certainty levels. Thin grey lines trace individual trends, illustrating the variance from the group average.

To investigate the impact of state curiosity, interest, as well as spatial and visual exploration on the precision of cognitive map formation, we devised a Bayesian multilevel model on the composite memory score and included both pre-room curiosity and post-room interest ratings, as well as path roaming entropy and head-direction roaming entropy as the predictors. Again, duration was included as a covariate. By centring these predictors around the individual mean and including their averages centred around the grand mean, this model allowed us to dissociate intra- from inter-individual effects on cognitive map formation. Critically, at the intra-individual level, the model revealed a positive association between cognitive map precision and pre-room curiosity (posterior mean of $\beta_{\text{curiosity}} = 0.066$, 93%-HPDI = [0.027, 0.10]), and a tentative negative association with post-room interest (posterior mean of $\beta_{\text{interest}} = -0.023$, 93%-HPDI = [−0.060, 0.014]) (Fig. 5B). In addition, cognitive map precision was positively associated with path roaming entropy (posterior mean of $\beta_{\text{pathR}} = 0.80$, 93%-HPDI = [0.0076, 1.59]). There was no evidence for an intra-individual relationship with head-direction roaming entropy (posterior mean of $\beta_{\text{headRE}} = -0.075$, 93%-HPDI = [−1.22, 1.12] (Fig. 5B). In contrast, at the inter-individual level, the analysis provided weak evidence for a positive association between cognitive map precision and average head-direction

**Fig. 5 | Memory test drawings and the influence of state curiosity ratings, interest ratings and exploration metrics on the precision of cognitive map formation.** **A** Memory test example drawings for the Library. The reference image in the top left shows the Library's detailed layout. The other three images are participants' recall drawings, each scored by two independent raters with averages provided. The scores represent four domains on a 1 (poor) to 5 (excellent) scale: Object Presence (OP) – inclusion of spatially defining objects; Spatial Distortion and Rotation of Features (SD) – accuracy in representing the size, shape, and orientation of objects and room features; Relative Positioning (RP) – the precision of objects and feature placement relative to one another; and Spatial Proportion (SP) – accuracy of the distances between objects and structural elements in the room. **B** Posterior distributions of predictor weights on the precision of cognitive map formation. Posterior distributions reveal the weights of pre-room curiosity and post-room interest ratings (left column), and path and head-direction roaming entropy (RE) (right columns), on the precision of overall cognitive map formation, which is measured by a composite score averaging the four domain scores above. The density of the shaded shapes indicates the posterior distributions, with the width of each shape reflecting the level of uncertainty in the weight estimates. Black dots mark the posterior mean weights, and horizontal bars indicate 67% (thick) and 93% (thin) HPDIs. Dashed vertical lines at zero indicate no effect on cognitive map formation.

## A. Example drawings

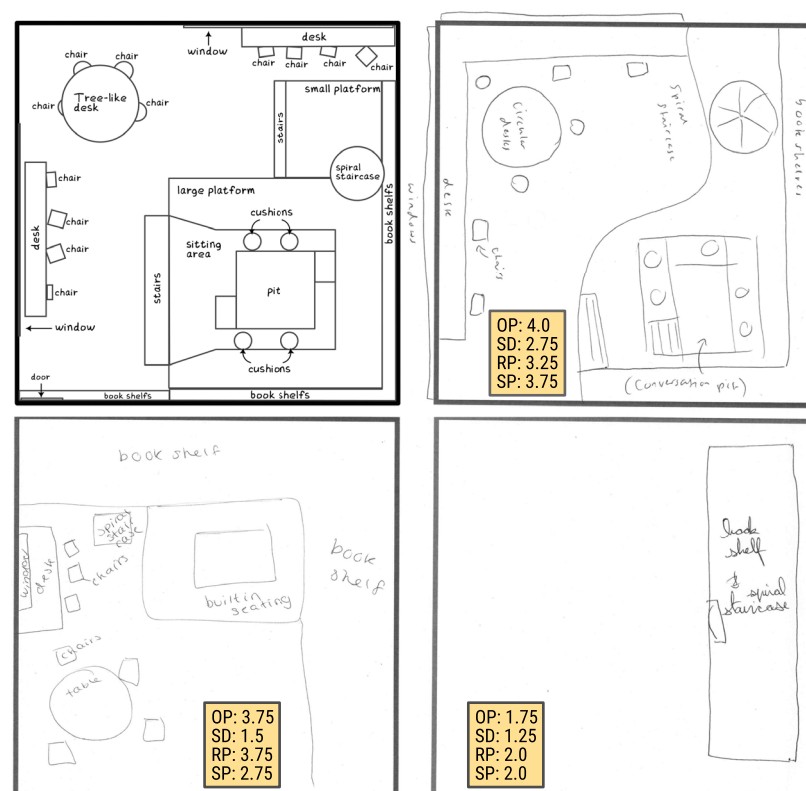

## B. Posterior distributions of predictor weights

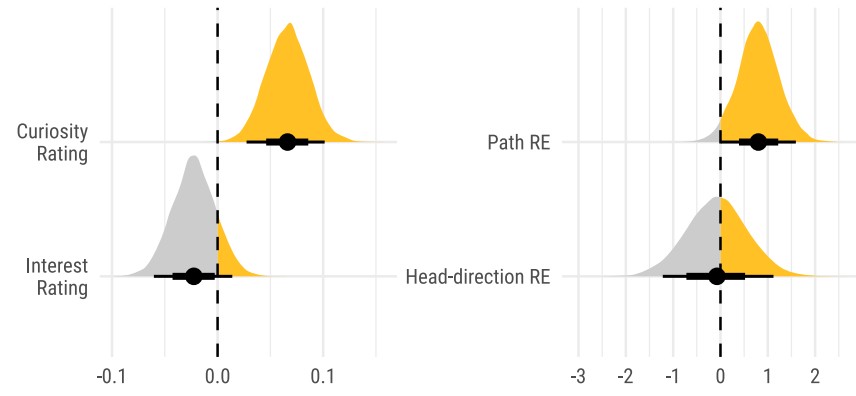

roaming entropy only (posterior mean of $\beta_{headRE_{avg}}$ = 2.22, 93%-HPDI = [−0.47, 4.80]). Importantly, our model indicated that the positive associations of cognitive map precision with intra-individual curiosity and path roaming entropy were not merely a byproduct of overall memory performance (see Supplementary Note 4 for further details).

In a follow-up analysis, we explored how the predictors, particularly pre-room curiosity and path roaming entropy, influenced each of the four cognitive map dimensions independently. Pre-room curiosity consistently enhanced all dimensions, indicating a broad positive impact on cognitive map formation. Path roaming entropy positively influenced Object Presence and may have potential benefits on Spatial Distortion and Rotation of Features, reflecting a more nuanced contribution to cognitive map formation (see Fig. S8 and Supplementary Note 5 for more details).

We further conducted a mediation analysis to ascertain if the effect of pre-room curiosity on cognitive map precision was mediated by path roaming entropy. The posterior mean for the mediation effect was estimated at 0.0028, with a 93%-HPDI spanning from –0.00035 to 0.0077 (see Fig. S9).

This result suggests that the influence of pre-room curiosity on cognitive map precision may be mediated by path roaming entropy. However, given the proximity of the posterior mean and HPDI to zero, any mediation effect, if existent, is likely to be modest.

## Discussion

It has long been thought that intrinsic states, such as curiosity, are critical for driving exploration in novel spatial environments and may thereby be important in forming unified representations of a place - or 'cognitive maps'[5–7]. We addressed this question in the current study and demonstrated a positive correlation between curiosity states and spatial exploration, as well as an enhancing effect of curiosity states on cognitive map formation in humans. Specifically, our findings reveal that an individual's pre-room curiosity about novel environments was positively associated with their level of spatial exploration (quantified by path roaming entropy). Furthermore, we found that individual differences, particularly in curiosity traits such as Stress Tolerance, modulate the relationship between curiosity and spatial

exploration. This suggests that the capacity to cope with the uncertainty of new situations is critical in enhancing the link between curiosity and exploratory behaviour. In contrast to the relationship between curiosity on spatial exploration, we found that the perceived interestingness of a room (rated after visiting a room) correlated with the degree of visual exploration (quantified by head-direction roaming entropy). Such a distinction unveils a double dissociation in how curiosity and interest are associated with different facets of spatial exploratory behaviour in humans.

Finally, we revealed that curiosity – alongside exploration—was strongly related to the accuracy and spatial-relational precision of individuals' memories, suggesting a fundamental role of curiosity-driven exploration in cognitive map formation[5-7]. This positive influence of curiosity on spatial exploration and memory aligns with theories of curiosity as a cognitive enhancer (e.g., refs. [6,9,11]) and extends previous research that has primarily focused on curiosity's role in enhancing memory for verbal and semantic information (e.g., refs. [14,15,30,55]). Our findings bridge these theoretical perspectives with spatial cognition, demonstrating curiosity's broader impact on cognitive processes.

Exploration is the process by which animals and humans gather information from novel surroundings[56-58], and is generally considered to be a form of spontaneous, self-paced behaviour in the absence of explicit reinforcers[7,58-60]. When arriving in a new environment, animals exhibit a sequence of excursions to inspect the surroundings, for example, locomotion into an approachable arena[57,59]. Prominent theories suggest that spatial exploration is stimulated by curiosity[5,61,62]. In practical terms, exploratory behaviours are taken as a behavioural expression of curiosity[63,64]. However, given the difficulty of manipulating and recording curiosity levels in animals, the hypothesis about the role of curiosity in underlying spatial exploration has not been tested behaviourally. Here, we showed that anticipation of novel rooms evoked various levels of curious states, leading to variation in the exploratory behaviour in humans that are in line with the observations of animals' exploratory behaviours in novel environments.

In addition to research on animal exploration, our results also align with several initial studies on how curiosity triggers human information-seeking and exploratory behaviour, and are the first in the domain of spatial exploration. These recent studies have shown that curiosity affects information-seeking and exploratory behaviours such as Wikipedia online search[18] and COVID-19-related information-seeking[13]. It has also been shown that curiosity states motivate anticipatory gaze towards the predicted position of upcoming curiosity-relieving information[65]. In contrast to animal research where spatial exploration has been widely investigated, there is little evidence from human studies. Our results extend prior research into the domain of spatial exploration and suggest that curiosity states influence spatial exploratory behaviours in humans, providing the first direct evidence (to our best knowledge) for the view that curiosity motivates spatial exploration.

By asking participants (in Experiment 2) to visually recall spatial environments which allowed different cognitive map properties to be coded reliably for individual environments, our results also suggested an important role of curiosity states in driving how precisely information was encoded into cognitive maps[6,66]. The observed positive association between pre-room curiosity and cognitive map formation, particularly at the intra-individual level, suggests that heightened curiosity may facilitate more precise cognitive map formation. Pre-room curiosity, as an anticipatory state, may activate broader cognitive schemas related to spatial navigation and environmental learning, facilitating the integration of new spatial information into existing frameworks. This builds upon an ever-growing body of literature suggesting that curiosity is important for item and verbal memory (e.g., answers to trivia questions[14,15]), but also critically extends this to the visual recall of complex relational information in the form of cognitive maps of spatial environments, further strengthening the view that curiosity modulates hippocampus-dependent memory representations[9].

Our findings also provide further support for a relationship between exploration and cognitive map formation and spatial learning[2,67-70]. Notably, unlike a recent study that measured exploration through a larger environment connected by designated paths[2], we found that path roaming entropy —defined as the degree of spatial coverage in an open environment - was related to the precision of cognitive maps. This difference may be attributed to our largely open, path-free environments that more closely mimic naturalistic settings, allowing for unrestricted exploration and potentially more thorough encoding of spatial relationships.

In terms of the potential neural mechanism underpinning the relationship between curiosity-based exploration and cognitive map formation, human studies have shown that a higher curiosity state is associated with increased activation in midbrain dopamine-related areas and the striatum[14,15,17,22,71,72]. Such increased activity within dopaminergic regions has been associated with increased hippocampus-dependent learning and memory consolidation[73-77]. Moreover, rodent studies have shown that dopamine stabilises cognitive map formation in the hippocampus[78-81], and theoretical models propose exploration of novel environments as a critical component of cognitive map formation in the hippocampus[82,83]. Therefore, both curiosity-enhanced memory *and* cognitive map formation via spatial exploration appear to rely on similar neural mechanisms, namely, the dopaminergic circuit and its interaction with the hippocampus[9,74,82-84]. It will be important for future work to further examine the possible role of dopaminergic-hippocampal interactions in curiosity-based exploration and thereby cognitive map formation in humans.

Another key finding of our study is the differential effects of states of curiosity and interest on spatial exploratory behaviours, suggesting distinct cognitive mechanisms. Curiosity is often operationalised through exploration and information-seeking behaviours aimed at resolving uncertainty, which ultimately enhances learning and cognitive performance e.g. refs. [8,9,11,12]. In contrast, post-room interest appears to engage more focused attention, driving visual exploration (head-direction roaming entropy) of specific features or objects. However, this interest-driven selective focus may occur at the expense of broader spatial encoding, which could potentially contribute to the observed negative trends in its association with path roaming entropy (Fig. 3) and cognitive map formation (Fig. 5B).

Our findings parallel observations in quadrupedal species, where behaviours like rearing in rodents - standing on their hindlimbs to gain an elevated perspective on a spatial environment[85] - are critical for mapping novel environments[6,86]. Our head-direction roaming entropy measure might capture analogue moments when participants paused to visually scan the environment, similar to rearing in animals, highlighting a distinct aspect of exploration. One possibility is that curiosity drives movement through space (path roaming entropy) to discover novel parts of an environment, while subjective feelings of interest may promote visual exploration (head-direction roaming entropy) to examine these discovered parts more closely. Moreover, as path roaming entropy was positively associated with cognitive map formation at the intra-individual level while head-direction roaming entropy showed a potential positive association at the inter-individual level, our findings further support the idea that path roaming entropy and head-direction roaming entropy are driven by distinct mechanisms. Recent work in rats[86] has shown that the hippocampus switches to a different processing mode during rearing episodes, with certain populations of hippocampal pyramidal cells exhibiting opposite firing patterns during movement versus rearing. Although neural mechanisms in humans may differ, this research underscores the importance of distinguishing between types of exploratory behaviour in spatial learning and memory formation.

Interestingly, while pre-room curiosity positively affected cognitive map formation, there was a potential negative effect of post-room interest on cognitive map formation. This finding contrasts with research using the trivia paradigm, where post-information interest typically enhances memory for specific items, such as trivia answers[10,29-32,87]. We speculate that in our spatial task, interest may enhance memory for specific objects or features, potentially at the expense of encoding the overall spatial layout crucial for cognitive map formation. While our study did not directly examine item memory, which could potentially be enhanced by post-room interest, this distinction emphasises the need for further research into how curiosity and interest differentially affect various types of memory formation in spatial learning contexts.

Notably, the effects that we have observed in this study reflected not only trial-by-trial variation in curiosity at the individual-subject level but also the impact of individual differences on curiosity-driven exploration at the trait level. Prior research by Risko and colleagues has shown a positive relationship between perceptual trait curiosity and visual exploration of a static scenery image[88]. Here, our findings demonstrated that individual differences in Stress Tolerance strengthen the positive association between curiosity and spatial exploration. Stress Tolerance has recently been suggested as one of the dimensions of trait curiosity, and has its importance in coping with uncertainty and engaging in processes that resolve uncertainty. This ability to handle anxiety arising from new situations is considered crucial in fostering curiosity and thereby promoting exploration to resolve uncertainty[35]. Our finding that individuals with higher Stress Tolerance are more likely to act upon their curiosity and therefore explore novel, uncertain environments is in line with recent findings that also showed that individual differences in Stress Tolerance were associated with more effective information sampling strategies[89] and more engagement in uncertainty-driven exploration[90]. Furthermore, the impact of Stress Tolerance in eliciting curiosity-based exploration informs recent theoretical ideas that speculate about the role of meta-cognitive components and appraisal to elicit curiosity-based learning[9,11,91–93]. Our findings on the impact of Stress Tolerance might indicate that meta-cognitive processes are not only essential for eliciting curiosity (as proposed by current theories) but also determine to which degree curiosity leads to exploratory behaviour.

## Limitations

Our study provides valuable insights into curiosity, exploration, and cognitive map formation, but several limitations warrant consideration. First, while our sketch map task is a validated measure of cognitive map accuracy, future studies could benefit from using complementary methods, such as navigation or wayfinding tasks, alongside the sketch map task. This would provide a more comprehensive assessment of cognitive map precision by capturing different aspects of spatial knowledge and its application. Second, while we interpret path roaming entropy primarily as a measure of spatial exploration, capturing the participant's movement through space, it is important to acknowledge that it may also reflect some aspects of visual exploration, as participants encounter new visual stimuli while moving. Similarly, head-direction roaming entropy predominantly measures visual exploration, tracking the orientation of the participant's head as they scan their environment. However, it may also be influenced by the spatial layout of the environment. Despite these overlaps, most of the variance in path roaming entropy and head-direction roaming entropy is expected to correspond to spatial and visual exploration, respectively. Third, both our experiments were heavily skewed toward women and future studies on curiosity-based exploration should aim for an equal gender balance to investigate whether the observed findings are generalisable across gender. Lastly, a detailed analysis of environmental features and their relationship to interest and exploration presents an exciting avenue for future research, potentially elucidating the complex interplay between interest, visual exploration and environmental characteristics.

## Conclusion

In conclusion, we found critical evidence of how curiosity is associated with spatial exploration that ultimately triggers the construction of cognitive maps – a central idea proposed by cognitive map theory[6] but that had never been tested. To the best of our knowledge, this is the first time that states of curiosity have been demonstrated to enhance spatial exploration and cognitive map formation. Understanding the influence of curiosity on human exploration has also practical implications for architecture, urban planning, museums, and game designs, which would benefit from harnessing curiosity to positively affect exploration and memory in real-world or virtual environments.

## Data availability

All data, on which this paper is based, are available at https://osf.io/sc37a.

## Code availability

Data were analysed with Python (3) and R (4.3.2) scripts. The code allowing to reproduce the presented analyses is available on the Open Science Framework (https://osf.io/sc37a/). The code for our online coding platform is available at https://osf.io/s2ja7.

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

## Acknowledgements
We thank Kathrin Eschmann, Rob Honey, and Ellen O'Donoghue for their comments on an earlier version and the revision of this manuscript. We also thank Yunxuan Ma and Sajida Yasin for their help with scoring the map drawings. For the purpose of Open Access, the authors have applied a CC-BY public copyright license to any Author Accepted Manuscript version arising from this submission. This work was supported by Wellcome Trust and Royal Society Sir Henry Dale Fellowship to M.J.G. (211201/Z/18/Z), the Biotechnology and Biological Sciences Research Council (BBSRC) Responsive Mode grant to C.J.H. (BB/V010549/1), and the Economic and Social Research Council (ESRC) Postdoctoral Fellowship to D.C. (ES/S011706/1). The funders had no role in study design, data collection and analysis, decision to publish or preparation of the manuscript.

## Author contributions
D.C., E.T., C.J.H., and M.J.G. contributed to the conceptualisation and methodology of the study. D.C. and E.T. carried out the investigation, with D.C. also responsible for visualisation. Funding acquisition was secured by D.C., C.J.H., and M.J.G. Project administration was managed by D.C. and M.J.G., under the supervision of M.J.G. The original draft was written by D.C., C.J.H., and M.J.G., and all authors (D.C., E.T., C.J.H., and M.J.G.) participated in the review and editing of the manuscript.

## Competing interests
The authors declare no competing interests.
