## [Peer Review File · Communications Psychology]

Curiosity shapes spatial exploration and cognitive map formation in humans

Corresponding Author: Dr Danlu Cen

Version 0:

Decision Letter:

Dear Dr Cen,

Thank you for your patience during the peer-review process. Your manuscript titled "Curiosity shapes spatial exploration and cognitive map formation in humans" has now been seen by 3 reviewers, and I include their comments at the end of this message. They find your work of interest but raised some important points. We are interested in the possibility of publishing your study in Communications Psychology, but would like to consider your responses to these concerns and assess a revised manuscript before we make a final decision on publication.

We therefore invite you to revise and resubmit your manuscript, along with a point-by-point response to the reviewers. Please highlight all changes in the manuscript text file.

Editorially, we consider particularly important that you clarify the statistical links between your variables (path-RE X head-RE; curiosity X interest; Reviewer#2). Your revision should also address potential confounders and mediators of your reported effects (Reviewer #1), and report on the variables that have been omitted (Reviewer #1, Reviewer #3). Finally, the referees' reports contain a number of questions and suggestions that should help you improve on the contextualization of your study, and on the interpretation of your results. Here, we highlight that speculation about neural underpinnings should be kept to an absolute minimum and clearly flagged as inferential, not demonstrated by the current data.

I am attaching an Editorial Requests Table that details critical reporting requirements for the revised manuscript. Please attend to each item and ensure your manuscript is fully compliant. We are requesting that your manuscript aligns with these requirements as this facilitates the evaluation of your manuscript, reducing delays in re-review and potential future acceptance. If your revised manuscript is not aligned with these requests on major issues, such as those concerning statistics, it may be returned to you for further revisions without re-review. Additional information can be found in our style and formatting guide <https://www.nature.com/documents/commspsychol-style-formatting-guide-accept.pdf>>Communications Psychology formatting guide.

Please use the following link to submit your

- revised manuscript,
- point-by-point response to the referees' comments,
- cover letter (as a separate document),
- the Editorial Policy Checklist (see below),
- the Reporting Summary (see below), and
- the completed Editorial Request Table (attached):

Link Redacted

Best regards,

Mael Lebreton

Mael Lebreton, PhD
Editorial Board Member
Communications Psychology
orcid.org/0000-0002-2071-4890

REVIEWER EXPERTISE:

Reviewer #1 curiosity; spatial navigation; cognitive maps

Reviewer #2 curiosity; decision-making

Reviewer #3 curiosity; spatial navigation; cognitive maps

REVIEWER REPORTS:

Reviewer #1 (Remarks to the Author):

This study aims to understand how curiosity and interest relate to exploration of novel environments. The authors use a virtual reality setup to characterise participants' patterns of movement and gaze when exploring rooms that they have not encountered before. They demonstrate a dissociation between spatial exploration (i.e., where participants moved) and visual exploration (where they looked), whereby spatial exploration was predicted by curiosity whereas visual exploration was predicted by reported interest in the environment.

Overall the paper is interesting and represents a well-conducted and creative piece of research. I do however have a few comments that could be addressed in a revision.

- The link to dopamine/hippocampus seems overstated in the case of this study and what it is able to test; this could be toned down slightly given that this study isn't directly investigating dopamine/hippocampal involvement, to avoid potentially misleading readers
- It would be worth defining curiosity more clearly, as it is mentioned a lot but not concretely defined
- Could the results be explained by some third variable that isn't investigated, such as the objects in the room? For example, if a room has lots of objects this may both make it more interesting and influence exploration, without interestingness necessarily predicting exploration
- It would be nice to see some information about the general interestingness and curiosity ratings for each of the rooms, just to give the reader an idea of what these looked like
- Could the process of asking people to rate (and consider) how curious they are have influenced their exploration behaviour? Perhaps if I think about how curious I am, it will affect my subsequent behaviour
- Bayesian modelling is an excellent choice of analysis for this data, however I have some concerns regarding the interpretation and reporting. The authors are in effect reporting the presence or absence of an effect, which relies on an (admittedly arbitrary) cutoff; they have chosen to use the 93% HPDI for this but it is not clear why they have chosen 93% rather than 95% which would be more conventional. It would be worth reporting 95% (unless they have a strong reason for using 93%). There is also a risk that to a reader who is less familiar with Bayesian statistics that this looks as though the goalposts are being moved in order to exaggerate the importance of effects.
- People with higher pre-level curiosity had better drawings/cognitive map – could it be these people were 1) overall more curious, 2) better at memory, and 3) better at drawing?
- Is it reasonable to conclude that being able to draw a map from memory implies that they have formed a cognitive map? This may be better tested using some task that relies on an actual cognitive map (e.g., navigation). A 2D, birds-eye view could just represent a visual representation of the room, without necessarily being dependent on an actual cognitive map of the spatial relations between places in the room. Has this approach been previously validated?
- It seems like only the drawing aspect of the memory test is reported. There are other aspects (e.g., object recognition, free recall) that do not seem to be analysed. It would be nice to see how these behaviours are influenced by curiosity and interest ratings. There is a brief line in the methods in relation to this, which doesn't seem to make sense: "The text responses of this free-recall test were used somewhere else".

Reviewer #2 (Remarks to the Author):

This is a very nice study on the relationship between subjective ratings of curiosity and interest and spatial exploration and memory. The authors designed a new task in which participants navigated through several virtual rooms, as well as reporting their curiosity (before exploring the room) and interest (after exploring the room), and completing the 5DC curiosity questionnaire and a surprise memory test for the room's layout after the exploration. The results show that higher pre-exploration curiosity was associated with a more distributed pattern of physical exploration (higher Roaming Entropy, RE), while post-exploration "interest" was associated with a more distributed pattern of visual exploration (higher head-direction RE). The former relationship between curiosity and RE was mediated by personality traits including thrill seeking and stress tolerance. Finally, RE, head-direction RE and curiosity had positive impacts on the quality of the participants' memory for the scenes, while interest had a more modest but still significant negative impact. The task is very useful and creative, the paper is very well written and the results provide new insights into the relationships between these fascinating constructs.

My comments consist mostly of requests for the authors to (1) better describe and motivate the measures they use, and (2) consider more deeply the explanations for the associations that they expect, or find among these measures.

1. The measures of path-RE and head-RE seem interesting, but the authors should provide more information about how they are distributed, how they are related and how they vary with the conditions of the task. pathRE may be positively correlated with headRE if participants predominantly view items by moving toward them. However, the two measures may also be negatively correlated if, for instance, a participant is in one place reducing pathRE, but turn around in that place, increasing headRE. This tradeoff – whether the measures are positively or negatively correlated – may in turn depend on the time available for exploration and the layout of the room. If one is in a hurry to get through a room (finish the task) or if one is in a room where most interesting landmarks can be seen from the entrance one may use less pathRE and more headRE. On the other hand, if one takes more time in a room, or if one is in a room where the most interesting items cannot be seen from one spot (e.g., the museum room) then one may achieve headRE mostly by increasing pathRE. These considerations bring about multiple questions that the authors should address in their revision: a) provide a description of the time that participants had – what were their instructions, how long did they spend, how this differed by room, etc.; b) better describe the visibility of objects in a room – to what extent interesting objects could or could not be seen from one spot?; c) show the distributions of pathRE and headRE measures and the correlations between them; d) discuss how the distributions/correlations in c depend on time spent and on the type of room.

2. It be useful to have additional insights about the measures of pre-exploration curiosity and post-exploration interest. The authors should better describe: a) how were the ratings distributed within and across participants; b) how did the two ratings relate to each other (was more curiosity associated with more or less interest?); c) related to b, how do the authors think these two measures differ?; d) was "interest" a prospective measure (i.e., "how interested would you be to learn more about that room?") or a retrospective one ("how interested were you while exploring the room?").

3. It would be very helpful to have a bit more principled explanations about the associations reported. Some explanations are provided – e.g., about how animals may structure pathRE and headRE during visual exploration, and about the role of stress tolerance in moderating the relation between curiosity and path RE -- and I found them very helpful in understanding the data. But I missed a similar understanding of the distinct roles of pathRE and headRE, or of curiosity and interest in spatial memory (or "cognitive maps"). How do you think each measure contributes? What explains the negative association between memory and interest? And what do you think is the direction of the relationship: does higher curiosity/pathRE/headRE promote better spatial memory, or does a better capacity for spatial memory promote higher curiosity/pathRE/headRE? I realize that a lot of this discussion would need to be speculative, but it is nevertheless important to understand whether and how these measures are important.

Minor: Several Methodological details should be clarified:

4. How did participants receive the 24-hour test in exp 1? Was this online or were they invited back in the lab?

5. Exploration speed was limited to 2 eye-heights/second. What is an "eye height"?

6. No results were reported from the incidental memory test – are these analyzed separately, or did they not produce significant findings?

Nice work!

Jackie Gottlieb

Reviewer #3 (Remarks to the Author):

Title : Curiosity shapes spatial exploration and cognitive map formation in humans

The purpose of this study is original in that it investigates the role of curiosity in spatial exploration and subsequent spatial memory. Indeed, on the one hand, the role of curiosity as a memory enhancer has already been shown (e.g., Gruber & Ranganath, 2019; Marvin & Shohamy, 2016; Murayama, 2022;), and on the other hand, curiosity states have been widely operationalized by exploration and information-seeking behaviors to resolve a lack of information in a given activity/task, or a situation of uncertainty in a given environment, and ultimately be rewarded with learning that increases us cognitively (e.g., Gottlieb & Oudeyer, 2018; Murayama, 2022; Wang & Hayden, 2021).

Here, the major contribution is to demonstrate the link between these two literatures, using virtual environments to show that aroused curiosity predicts the amount of spatial exploration, and that curiosity and exploration are independently beneficial for spatial memory (learning a cognitive map of the environment). In this way, the initial intuitions of historical contributors such as Tolman (1948) or O'Keefe & Nadel (1978) to the study of spatial exploration and spatial memory in unfamiliar environments, i.e., curiosity, novelty seeking could be primary motives for exploration behaviors then requiring orientation and spatial memory skills.

Apart from this significant contribution, the other contribution is to offer a new VR-based experimental paradigm for exploring the links between curiosity and memory. Today, most studies only use verbal material (i.e., Trivia paradigm). Using this type of paradigm, it will be possible to study and even generalize the results obtained with verbal stimuli to non-verbal stimuli, such as visuospatial information. In addition, the strong positive relation between curiosity estimates and Roaming Entropy (RE) measures provides a promising way for probing curiosity states with objective measurements rather than subjective ones based on self-rating methods (that exhibit some limitations, e.g., Fastrich, G. M., Kerr, T., Castel, A. D., and Murayama, K. (2018). The role of interest in memory for trivia questions: an investigation with a large-scale database. *Motiv. Sci.* 4, 227–250. doi: 10.1037/mot0000087).

All in all, it's clear that this study is convincing and, in my expert opinion, deserves to be read and shared with readers of *Nature -Psychology* communication. However, I do have a few comments and points I'd like the authors to clarify, as follows.

1) Regarding the state of the art :

- p. 2. It is unclear why the authors contrast semantic knowledge acquisition and spatial exploration behaviors. Semantic knowledge has been shown to be extracted from curious spatial exploration. (Jirout, J., and Klahr, D. (2012). Children's scientific curiosity: In search of an operational definition of an elusive concept. *Dev. Rev.* 32, 125–160. doi: 10.1016/j.dr.2012.04.002 ; Jirout, J. J. (2020). Supporting early scientific thinking through curiosity. *Front. Psychol.* 11:1717. doi: 10.3389/fpsyg.2020.01717). For their study purpose, it's more the nature of the stimuli (verbal vs. non-verbal) that seems to matter, because in the end, curiosity-driven spatial exploration always aims for either perceptual and/or semantic stimulation. Moreover, there is a recent study with children revealing the benefit from curiosity on spatial memory (Sivashankar, Y., Fernandes, M., Oudeyer, P. Y., & Sauz on, H. (2024). The beneficial role of curiosity on route memory in children. *Frontiers in Cognition*, 3.)

- p. 2 -3 : The authors refer extensively to their own and related work in the field of functional neuroscience, but work in the psychology of curiosity is worth mentioning, particularly to clarify the distinctions between perceptual vs. epistemic curiosity (e.g., Berlyne 1954, Litman, 2000, Collins, R. P., Litman, & Spielberger, 2004). The measurement of perceptual curiosity. *Personality and individual differences*, 36(5), 1127-1141). It would also be helpful if the authors could clearly define what is meant by curiosity, interest and novelty. At times, one gets the impression that curiosity and novelty are equated, which is confusing (cf. Murayama et al., 2019). Sometimes, spatial and visual explorations are distinguished as if path-RE refers to spatial exploration while Head-RE refers to visual exploration. Definitions of distinctions and terminologies could prove useful in interpreting the double empirical dissociation reported in this study.

2) Regarding the study design :

-Two studies are proposed using the same virtual application in which the participant freely explores a series of virtual rooms (n=16), and from which are measured:

- the subjective state of curiosity before and after exposure to the virtual room
- the participants' exploratory behaviour in each room is probed by Roaming Entropy (RE) measures (Freund et al., 2013) whose increase in value indicates a broad, or complex, exploration of the environment. Two RE measures are calculated: i) "Path RE" from the translational moves (via clicks of "W" and "S" letters on keyboard forwards/ backwards, respectively) for estimating the explored surface by the participant for each room (location data), ii) "head direction RE" from directional moves (via the tracking of the computer mouse changes in the direction of the movement and in the angle of the frame/field of view) for probing head direction moves
- the memory performance of the path performed in each "room" is measured with a drawing task associated either with a yes/no recognition for the Exp. 1 (n =28) or with a free recall for the Exp. 2 (N=60). For the latter, the impact of trait curiosity (measured with the scale of Kashdan et al., 2020) is proposed. Why take a dimension out of the test and risk impoverishing its psychometric quality? If, as the authors presume, the excluded dimension is far removed from the construct of curiosity targeted in spatial exploration, it can serve as a measure of divergent validity. Also, for exp. 2., a new method of coding spatial memory performance is created to evaluate 4 dimensions in the formation of a cognitive map, i.e., Object Presence, Spatial Distortion and Rotation of Features, Relative Positioning, and Spatial Proportions. why only the structuring objects in the room and not also the other ones, that can have attractiveness value for a deeply exploration through curiosity?
- Overall, the changes in methods across the 2 experiments are rather confusing, and one comes to think that Exp. 1 is a pilot study and Exp. 2 the real study; so does Exp. 1 belong in the core of the paper, shouldn't it be in the supplementary material? This question is also motivated by the results that are reported as more reliable for Exp. 2 compared to those from Exp. 1, especially regarding the dissociation effect.
- The main hypotheses are: "i) the high states of pre-room curiosity positively affect the magnitude of spatial exploration while ii) the states of post-room curiosity are expected as positively related to both path-RE and head-direction RE.

3) Regarding the results :

- The statistical treatments are well documented, and based on Bayesian models. The exploration duration is critically controlled in the correlations.
- As previously mentioned, the core of the main document may include only the more reliable and well-built results, i.e. results from the Exp. 2.
- A double dissociation between the effects of pre-room curiosity and post-room interest is observed on the two types of explorative behaviours : 1) there is a positive correlation between pre-room curiosity and path RE that is moderated by individual differences in trait curiosity (especially for individuals with high Stress Tolerance and high Deprivation Sensitivity; 2) There is a positive association between post-room interest and Head-direction RE
- Cognitive map precision is positively related with pre-room curiosity and the two RE scores while it is negatively related with post-room interest
- Pre-room curiosity has a positive impact on all the five dimensions of cognitive map while Path RE positively influenced only object Presence of cognitive map formation.

4) Regarding the discussion/conclusion :

- a double dissociation is shown, but at the end we wonder to which cognitive dimensions of curiosity this corresponds? Once again, neurofunctional models are taken up, but they don't shed any light on what could correspond to the priori curiosity - Path RE coupling, and the a posteriori curiosity - Head directions RE couplings. Psychological studies could help with interpretation but so could the path-integration model (e.g., Wolbers & Hegarty, 2010; Anastasiou, C., Baumann, O., & Yamamoto, N. (2023). Does path integration contribute to human navigation in large-scale space?. *Psychonomic Bulletin & Review*, 30(3), 822-842.)
- In this study, room navigation is based on movement metaphors (keyboard for translational movement and mouse for directional movement), so movement is less embodied and mediated by repetitive "cognitive translations" of the metaphors. As a result, this may have reduced the sensory integration of the path, and hence of the cognitive maps constructed by the participants. Another consequence is that the use of metaphors may have reduced the sense of presence in virtual environments, and been deleterious to post-visit curiosity, thus disrupting cognitive map formation. In this respect, it would be interesting to have the differences between pre- and post-visit curiosity assessments, to see if the way of navigating the VR application had any consequences on subsequent curiosity states. Maybe, it is a possible explanation that the RE did not influence the 5 dimensions of cognitive map?
- The authors, in addition to highlighting their contributions, should also highlight the limits to be set in the future to support their (interesting) contribution.

EDITORIAL POLICIES

We ask that you ensure your manuscript complies with our editorial policies and reporting requirements.

To that end, we require revised manuscripts to be accompanied by two completed items: a reporting summary that collects information on study design and procedure, and an editorial policy checklist that verifies compliance with all required editorial policies.

- <https://www.nature.com/documents/nr-reporting-summary.zip>>Nature Research Reporting Summary
- <https://www.nature.com/documents/nr-editorial-policy-checklist.pdf>>Editorial Policy Checklist

All points on the policy checklist must be addressed. Your revised manuscript can only be sent back to the referees if these checklists are completed and uploaded with the revision.

Notes: If you have submitted a Stage 1 Registered Report, Review, Primer, Comment, or Perspective you do not need to submit these forms. If you have already submitted these forms, you may disregard this request.

** Visit Nature Research's author and referees' website at <http://www.nature.com/authors>>www.nature.com/authors for information about policies, services and author benefits**

Version 1:

Decision Letter:

Dear Dr Cen,

Your manuscript titled "Curiosity shapes spatial exploration and cognitive map formation in humans" has now been seen by our reviewers, whose comments appear below. In light of their advice I am delighted to say that we are happy, in principle, to publish a suitably revised version in Communications Psychology.

We therefore invite you to revise your paper one last time to address the remaining concerns of our reviewers and a list of editorial requests. At the same time we ask that you edit your manuscript to comply with our format requirements and to maximise the accessibility and therefore the impact of your work.

EDITORIAL REQUESTS:

SUBMISSION INFORMATION:

In order to accept your paper, we require the list of files specified at <https://www.nature.com/documents/commsj-file-checklist.pdf>.

OPEN ACCESS:

*** TRANSPARENT PEER REVIEW:** Communications Psychology uses a transparent peer review system. On author request, confidential information and data can be removed from the published reviewer reports and rebuttal letters prior to publication. If you are concerned about the release of confidential data, please let us know specifically what information you would like to have removed. Please note that we cannot incorporate redactions for any other reasons.

Link Redacted

Best regards,

Marika

on behalf of

Mael Lebreton

Marika Schiffer, PhD
Chief Editor
Communications Psychology

Mael Lebreton, PhD
Editorial Board Member
Communications Psychology
orcid.org/0000-0002-2071-4890

REVIEWERS' COMMENTS:

Reviewer #1 (Remarks to the Author):

The authors have done a thorough job of addressing my comments and those of the other reviews, and the manuscript is much improved as a result. I am happy to recommend publication.

Reviewer #2 (Remarks to the Author):

This revision comprehensively addresses my previous concerns. The added information about methodology, room layouts, the distributions of different measures, and the interpretation is detailed and helpful, and make this paper a strong and interesting addition to the literature. I have only a few remaining minor comments:

1. The participant samples are heavily biased toward females. In exp. 1, 25 of 28 participants are female and in exp. 2, 55 of 60 participants are! I apologize for not noticing this in the first pass, but the bias is quite extreme and should be acknowledged. I suggest that the authors include a note about it in the section on "Limitations", and mention whether they have any reason to expect (or rule out) that gender/sex would affect the behaviors they study.
2. I appreciate the discussion of exploration duration, and I would like to ask for a bit more details on it. What did participants know at the start of the experiment about the rooms that were available to explore and the expected length of the experiment? E.g., were people expecting a 1 or 1.5 hour experiment, and were they told the total number of available rooms and/or the types of the rooms they would be able to access? Also, were participants required to explore all the rooms – and if not, did they explore all the rooms or did some people skip some rooms? If the latter, did people know the types of rooms they would have (and thus could skip some rooms strategically?) or were skipping decisions made on the spot (or by chance?). These details will help readers understand how people decided how much time to spend in each room.
3. The strong relation with Stress Tolerance is very intriguing, particularly since this personality measure shows up repeatedly in very different measures of curiosity and information gathering. I appreciate the citation of Rischall et al., which also finds Stress Tolerance to be an important predictor after exploring a wider range of personality traits. I wanted to alert the authors to a new paper that validates this result for a wider set of tasks:
<https://motivationsciencelab.com/publication/curiosity-in-cognitive-science-and-personality-psychology-individual-differences-in-information-demand-have-a-low-dimensional-structure-that-is-predicted-by-personality-traits/> The paper is in press at PNAS; there is no obligation to cite it, but I wanted to mention it in case the authors find it interesting to consider and/or comment on the relationship with their paper.

Thank you again for a comprehensive revision and congratulations on an excellent paper!
Jackie Gottlieb

Reviewer #3 (Remarks to the Author):

It's a version that satisfies my comments and the authors have modified or justified their answers. It's a great contribution to the field of curiosity!

Reviewer 1

This study aims to understand how curiosity and interest relate to exploration of novel environments. The authors use a virtual reality setup to characterise participants' patterns of movement and gaze when exploring rooms that they have not encountered before. They demonstrate a dissociation between spatial exploration (i.e., where participants moved) and visual exploration (where they looked), whereby spatial exploration was predicted by curiosity whereas visual exploration was predicted by reported interest in the environment.

Overall, the paper is interesting and represents a well-conducted and creative piece of research.

Response: We thank the reviewer for their positive comments about our study.

1. ** The link to dopamine/hippocampus seems overstated in the case of this study and what it is able to test; this could be toned down slightly given that this study isn't directly investigating dopamine/hippocampal involvement, to avoid potentially misleading readers.*

Response: The reviewer is correct that our study did not directly investigate the neural mechanisms underlying curiosity and cognitive map formation. Therefore, we have toned down these statements to accurately reflect the scope of our findings. In the Introduction, we have now deleted any references to dopamine/hippocampus to avoid misleading the readers. In the Discussion, there is now only one single paragraph that speculates about the possible role of dopamine/hippocampus in curiosity-related exploration and cognitive map formation.

Below is the revised paragraph from the Discussion section:

“In terms of the potential neural mechanism underpinning the relationship between curiosity-based exploration and cognitive map formation, human studies have shown that a higher curiosity state is associated with increased activation in midbrain dopamine-related areas and the striatum (Gruber et al., 2014; Jepma et al., 2012; Kang et al., 2009; Lau et al., 2020; Oosterwijk et al., 2020; Poh et al., 2022). Such increased activity within dopaminergic regions has been associated with increased hippocampus-dependent learning and memory consolidation (Lisman et al., 2011; Lisman & Grace, 2005; Murayama & Kitagami, 2014; Patil et al., 2017; Shohamy & Adcock, 2010). Moreover, rodent studies have shown that dopamine stabilises cognitive map formation in the hippocampus (DeYoung, 2013; Gomperts et al., 2015; Lansink et al., 2009; McNamara et al., 2014), and theoretical models propose exploration of novel environments as a critical component of cognitive map formation in the hippocampus (Düzel et al., 2010; Voss et al., 2017). Therefore, both curiosity-enhanced memory and cognitive map formation via spatial exploration appear to rely on similar neural mechanisms, namely, the dopaminergic circuit and its interaction with the hippocampus

(Düzel et al., 2010; Gruber & Ranganath, 2019; Lisman & Grace, 2005; Schomaker, 2019; Voss et al., 2017). It will be important for future work to further examine the possible role of dopaminergic-hippocampal interactions in curiosity-based exploration and thereby cognitive map formation in humans.” (p. 22, paragraph 2)

2. *It would be worth defining curiosity more clearly, as it is mentioned a lot but not concretely defined.*

Response: We thank the reviewer for highlighting the need for a clear definition of curiosity. Curiosity is indeed a multifaceted construct, and providing our definition is crucial for understanding its role in our study. In the revised manuscript, we now state:

“Early influential theories suggest that curiosity – the innate desire to seek out novel information – may be one of the primary drivers of exploratory behaviour” (p. 2, paragraph 1).

3. ** Could the results be explained by some third variable that isn’t investigated, such as the objects in the room? For example, if a room has lots of objects this may both make it more interesting and influence exploration, without interestingness necessarily predicting exploration.*

Response: We appreciate the reviewer’s comment regarding the potential influence of third variables, such as the number and nature of objects in the room, on our results. First, we should state that our key experimental result – the three-way relationship between curiosity, exploration, and memory – cannot be explained by number of objects in the room, as these are, critically, unknown to the participant when rating their curiosity (i.e., in the virtual walkway prior to entering the rooms).

In terms of the interest ratings, which were acquired *after* the exploration phase, we acknowledge that environmental factors (such as the number of objects in the room) could impact visual exploration (i.e., head-direction RE), and in turn, feelings of interestingness. To directly address this possibility, we included room type as a factor in our model. After controlling for room type, the positive relationship between interest ratings and exploration behaviour remained in both experiments. In Experiment 1, the wide distribution could be due to the smaller sample size. However, the association was positive in Experiment 2, and when we combined the data from both experiments, the relationship remained robust. Further details of this analysis are provided in Supplementary Note 3 and Fig. S7 (see below).

4. *It would be nice to see some information about the general interestingness and curiosity ratings for each of the rooms, just to give the reader an idea of what these looked like.*

Response: We have now included a figure with distributions of curiosity and interest ratings for both Experiment 1 and 2 in Supplementary Figure 4 in the supplementary information file (Fig. S4). The distributions of ratings for each room shows that the rooms elicited the full range of curiosity and interestingness ratings.

5. *Could the process of asking people to rate (and consider) how curious they are have influenced their exploration behaviour? Perhaps if I think about how curious I am, it will affect my subsequent behaviour.*

Response: This is an interesting point from the reviewer. Of course, for the purpose of testing our key hypothesis, we had to directly probe feelings of curiosity, though it would be an interesting follow-up investigation (with real-world implications) to compare explicit ratings of curiosity to more 'objective', or implicit, measures, such as willingness-to-wait. Indeed, studies that use subjective curiosity ratings alongside objective indices of curiosity (e.g., willingness to wait) show that they have similar influences on behaviour (e.g., Kang et al., 2009; van Lieshout et al., 2018) and correlate highly (e.g., Marvin & Shohamy, 2016).

Further, we did introduce a delay between the curiosity rating and the exploration phase (mean = 26.82 s, SD = 3.52 s) across our experiments. This temporal separation likely minimised any immediate influence of the rating task on subsequent behaviour.

6. *Bayesian modelling is an excellent choice of analysis for this data, however I have some concerns regarding the interpretation and reporting. The authors are in effect reporting the presence or absence of an effect, which relies on an (admittedly arbitrary) cutoff; they have chosen to use the 93% HPDI for this but it is not clear why they have chosen 93% rather than 95% which would be more conventional. It would be worth reporting 95% (unless they have a strong reason for using 93%). There is also a risk that to a reader who is less familiar with Bayesian statistics that this looks as though the goalposts are being moved in order to exaggerate the importance of effects.*

Response: We appreciate the reviewer's feedback regarding our use of the 93% HPDI and its interpretation. We would like to clarify that our intention in reporting the 93% HPDI was not to establish a cutoff for determining the presence or absence of an effect. Instead, it is intended to summarise the posterior distribution, which could be any interval as long as it helps in illustrating the shape of the posterior. Indeed, using arbitrary cutoffs (whether it be 93% or 95%) can be misleading. As the reviewer seems to be aware, instead of testing a hypothesis at an arbitrary level, Bayesian inference focuses on the probability of the hypothesis given the data and prior information. The use of cutoffs can oversimplify the results and ignore the continuous nature of probability that Bayesian methods provide.

As Richard McElreath (2020) emphasises in "Statistical Rethinking: A Bayesian Course with Examples in R and Stan (Second Edition)", the Bayesian parameter estimate is precisely the entire posterior distribution. This distribution maps each unique parameter value onto a plausibility value, summarising the relative plausibilities of each possible value. Intervals are merely helpful summaries, and if the choice of interval leads to different inferences, it is better to plot the entire posterior distribution. This is what we did in our results figures in which we plotted the entire posterior distribution for the parameters.

Following the suggestions by Richard McElreath (2020), we intentionally avoided the conventional 95% intervals to prevent readers from unconsciously conducting hypothesis tests. Conventional 95% intervals can encourage a binary interpretation, which is contrary to the Bayesian approach that focuses on updating beliefs based on the entire posterior distribution.

In the main text, we tried to emphasise a probabilistic interpretation of parameters when reporting the results. Take the relationship between path RE and post-room interest ratings for example, the posterior mean for the parameter estimate was -0.0040 and 93% HPDI was $[-0.0092, 0.0012]$. Based on this result, we inferred that there was a negative relationship between path RE and post-room interest ratings, but given the mass of the plausibilities of estimates being positive, we considered this relationship to

be potentially negative (page 12, last paragraph). In the Methods, we added: “We used posterior means and 93% Highest Posterior Density Intervals (HPDIs) to summarise the effects of predictors. The choice of 93% HPDI is meant to provide a summary of the distribution rather than a threshold for hypothesis testing. To present a comprehensive view of the data, we plotted the entire posterior distributions for parameters of interest, illustrating the relative plausibilities of each parameter value.” (p. 11, paragraph 4).

We hope these revisions address the reviewer’s concerns and provide a clearer understanding of our Bayesian analysis approach.

7. *People with higher pre-level curiosity had better drawings/cognitive map – could it be these people were 1) overall more curious, 2) better at memory, and 3) better at drawing?*

Response: We appreciate this comment about the potential influence of inter-individual differences in overall curiosity trait, overall memory, and drawing ability affecting our findings. In response, we have revised our analysis by centring the predictors (pre-room curiosity, post-room interest, path RE and head-direction RE) around their individual means and including their average values in the model. This allows us to separate intra-individual from inter-individual effects, which is critical as our main analyses focused on how varying states of curiosity between individual rooms (i.e., *intra*-individual changes in curiosity) affect cognitive map formation. In terms of inter-individual differences, as suggested by the review, we find no evidence that inter-individual differences in overall curiosity (e.g., mean levels of curiosity across the task) relate to memory overall (e.g., averaged across all drawings) (see the figure below and Supplementary Note 4 for details).

To investigate whether “overall memory” ability (i.e., inter-individual differences) explained the intra-individual effects of curiosity states on cognitive map formation, we examined the relationship between participants’ baseline memory abilities (random intercept) and the effect of curiosity states (random slope) in our multilevel model on composite memory score. The correlation between the random intercept and the curiosity slope was negative (posterior mean of correlation = -0.36, 93%-HPDI = [-0.79, 0.063]). This suggests that participants with higher baseline memory scores tend to show a weaker relationship between curiosity and memory performance. Therefore, the effect of curiosity on performance is unlikely to be driven by inherent memory abilities. In fact, curiosity appears to have a stronger effect on performance for participants with lower baseline memory scores. Further details have been added to Supplementary Note 5. Given that these findings could be potentially driven by a ceiling effect (i.e., participants with higher overall memory show less curiosity-related effects on memory), we refrain from adding this analysis to the main text.

In terms of drawing ability, we made sure to standardise the instructions and provided participants with templates to ensure uniformity in map drawing. More importantly, maps were scored based on predefined criteria for accuracy that focused on spatial relationships and room layout, for example, the correct identification and placement of key objects within the room and the spatial relationships between these objects. By emphasising these aspects, the scoring system minimised the influence of drawing ability and focused on the cognitive representation of the spatial environment. We have included this information in the Methods (under “Scoring of sketch maps”, p. 9).

8. *Is it reasonable to conclude that being able to draw a map from memory implies that they have formed a cognitive map? This may be better tested using some task that relies on an actual cognitive map (e.g., navigation). A 2D, birds-eye view could just represent a visual representation of the room, without necessarily being dependent on an actual cognitive map of the spatial relations between places in the room. Has this approach been previously validated?*

Response: We appreciate the reviewer’s comment on the validity of using a 2D, birds-eye view drawing task to infer cognitive map formation. We agree that this is an important consideration and would like to address it based on established research and the specifics of our study design.

Cognitive maps are internal, allocentric representations of spatial environments that enable the recovery of distances and directions between locations and flexible route planning (Gallistel, 1989; O’Keefe & Nadel, 1978). Our use of sketch maps to assess these cognitive maps is grounded in this theoretical framework and supported by extensive psychological research, dating back to Lynch (1960). Numerous studies have demonstrated that map drawing is an effective and reliable measure of spatial memory

(Billingshurst & Weghorst, 1995; Blades, 1990; Newcombe, 1985; Tversky, 1981) and a predictor of wayfinding performance (Rovine & Weisman, 1989).

In our study, participants explored virtual rooms from a first-person perspective (i.e. egocentric). Importantly, we designed the rooms so that they could not be fully seen from a single vantage point. This design ensured that high performance on our memory measures necessitated thorough navigation and the formation of a comprehensive cognitive map, rather than reliance on a single visual snapshot. The subsequent memory test, which required participants to draw 2D maps of the room layouts, specifically tapped into these allocentric representations.

It's important to note that these 2D birds-eye sketch maps are not merely visual representations of the rooms. Participants were explicitly instructed to depict spatial relations between objects and room features, requiring them to recall and accurately represent spatial relationships - a key component of cognitive maps. By asking participants to translate their first-person experiences into an overhead view, we assess their ability to manipulate and externalise their internal spatial representations. The accuracy of these sketch maps thus provides insight into how well participants have encoded the spatial layout of the rooms, reflecting a hallmark of cognitive map formation.

We have revised the manuscript to include a more detailed explanation of the rationale behind using the sketch map task, along with references to previous research validating this approach:

“Critically, Experiment 2 included a memory test in which participants were required to draw a map of the layout of each room (see Fig. 1A, 5A). Importantly, map drawings allow more insight into the precision and content of memory representations (Bainbridge, 2022). Building on previous measures for sketch map drawings (Billingshurst & Weghorst, 1995; Blades, 1990; Newcombe, 1985; Rovine & Weisman, 1989; Tversky, 1981), our novel coding scheme to these drawings (see Fig. 5A and methods in the supplementary materials) allowed us to assess how curiosity states enhance cognitive map formation of novel rooms.” (p. 3, last paragraph)

While the sketch map task is suitable for the current study, we acknowledge that tasks such as navigation or route planning could provide complementary insights into cognitive map formation. Future research could incorporate these methods to further validate and expand our understanding of cognitive map development in different contexts: *“First, while our sketch map task is a validated measure of cognitive map accuracy, future studies could benefit from using complementary methods, such as navigation or wayfinding tasks, alongside the sketch map task. This would provide a more comprehensive assessment of cognitive map precision by capturing different aspects of spatial knowledge and its application.” (p. 24, paragraph 2)*

9. ** It seems like only the drawing aspect of the memory test is reported. There are other aspects (e.g., object recognition, free recall) that do not seem to be analysed. It would be nice to see how these behaviours are influenced by curiosity and interest ratings. There is a brief line in the methods in relation to this, which doesn't seem to make sense: "The text responses of this free-recall test were used somewhere else".*

Response: We apologise for any confusion caused with regards to our primary measures in this study. These tests are not relevant to the hypotheses tested in the current study but the methods were left in the paper to ensure transparency of our procedure (which formed part of a larger investigation).

Experiment 1 included a memory test for the incidentally encountered objects on the pathway leading to the rooms. Experiment 2 included three memory tests: the sketch map task (reported here), free recall and introspective recall. Each test was designed to assess different aspects of memory and introspective experiences, which constitute different research questions.

Here, our primary goal was to investigate the relationship between curiosity, spatial exploration, and the formation of cognitive maps. Consequently, we focused on the sketch map task, which directly measures the accuracy of cognitive maps and aligns with our primary research objective. While verbal recall and introspective experiences is useful to examine object memory and emotional content of exploration, they are less useful for assessing precision of visuospatial representations. Including them would have shifted the focus from our core research question. Therefore, to maintain methodological consistency and clarity, we chose to prioritise spatial memory measures that directly addressed our primary hypotheses.

To avoid confusing the readers about additional memory tests (different memory tests across the two experiments), we decided to move all explanations of additional memory tests into the Supplemental Methods (also to ensure transparency of our approach): *"Although other memory tests were included in both Experiment 1 and 2 (see Supplementary Methods for details of those tests), these were designed to address independent research questions and will not be reported here."* (p. 5, paragraph 2)

Reviewer 2

This is a very nice study on the relationship between subjective ratings of curiosity and interest and spatial exploration and memory. The authors designed a new task in which participants navigated through several virtual rooms, as well as reporting their curiosity (before exploring the room) and interest (after exploring the room), and completing the 5DC curiosity questionnaire and a surprise memory test for the room's layout after the exploration. The results show that higher pre-exploration curiosity was associated with a more distributed pattern of physical exploration (higher Roaming Entropy, RE), while post-exploration "interest" was associated with

a more distributed pattern of visual exploration (higher head-direction RE). The former relationship between curiosity and RE was mediated by personality traits including thrill seeking and stress tolerance. Finally, RE, head-direction RE and curiosity had positive impacts on the quality of the participants' memory for the scenes, while interest had a more modest but still significant negative impact. The task is very useful and creative, the paper is very well written and the results provide new insights into the relationships between these fascinating constructs.

My comments consist mostly of requests for the authors to (1) better describe and motivate the measures they use, and (2) consider more deeply the explanations for the associations that they expect, or find among these measures.

Response: We thank the reviewer for their thoughtful review and positive assessment of our manuscript. We address all comments in detail below.

1. ** The measures of path-RE and head-RE seem interesting, but the authors should provide more information about how they are distributed, how they are related and how they vary with the conditions of the task. pathRE may be positively correlated with headRE if participants predominantly view items by moving toward them. However, the two measures may also be negatively correlated if, for instance, a participant s in one place reducing pathRE, but turn around in that place, increasing headRE. This tradeoff – whether the measures are positively or negatively correlated – may in turn depend on the time available for exploration and the layout of the room. If one is in a hurry to get through a room (finish the task) or if one is in a room where most interesting landmarks can be seen from the entrance one may use less pathRE and more headRE. On the other hand, if one takes more time in a room, or if one is in a room where the most interesting items cannot be seen from one spot (e.g., the museum room) then one may achieve headRE mostly by increasing pathRE. These considerations bring about multiple questions that the authors should address in their revision: a) provide a description of the time that participants had – what were their instructions, how long did they spend, how this differed by room, etc.; b) better describe the visibility of objects in a room – to what extent interesting objects could or could not be seen from one spot?; c) show the distributions of pathRE and headRE measures and the correlations between them; d) discuss how the distributions/correlations in c depend on time spent and on the type of room.*

Response: We appreciate the opportunity to provide further clarification, and details regarding our measures of Path RE and Head-direction RE. Below, we address each of the reviewer's specific points (a-d):

a) Provide a description of the time that participants had – what were their instructions, how long did they spend, how this differed by room, etc.

Participants were instructed to explore each room freely without any time constraints:

“Once inside, participants were instructed to explore each room freely without time constraints, allowing them to engage with the environment at their own pace.” (p. 5, paragraph 4). This instruction was intended to allow participants to engage with the environment at their own pace, thereby capturing naturalistic exploration behaviours.

On average, participants spent approximately 30 seconds in each room (Experiment 1: $M = 30.4$ s, $SD = 14.6$ s; Experiment 2: $M = 33.1$ s, $SD = 16.6$ s; as reported in the original manuscript), with variability observed across different rooms. In addition to the average durations for both experiments in the original manuscript (p. 12 – p. 13), we now added a supplementary figure in the supplementary information file (Fig. S6) to show the different distributions of time length for the rooms in both experiments.

b) Better describe the visibility of objects in a room – to what extent interesting objects could or could not be seen from one spot?

In our study, each room was designed to simulate a realistic environment with objects placed at varying degrees of visibility and accessibility to encourage exploration. Objects were distributed throughout the rooms such that most of them were in plain sight from multiple vantage points, while others were partially obscured or could only be seen from specific angles or upon closer inspection.

To address this, we revised the manuscript to include a more detailed description of the visibility and accessibility of objects in each room. Specifically, we described:

1. **Room layout and object placement:** In addition to the screenshots of the rooms (Fig. S1), we also provided a detailed layout of each room, highlighting the positions of key objects and furniture (Fig. S2) in the supplementary information file.

2. **Visibility conditions:** An explanation of how the room design influenced object visibility. For example, some objects were placed behind larger furniture pieces or inside cabinets, making them less visible from certain angles.

In the Methods section, under “Virtual environments and design”, we included: *“Each room was designed to promote naturalistic exploration, featuring a combination of easily visible layout-defining objects (e.g., sofa, bookshelf) and smaller details (e.g., plate on table, books on lower shelf). To further encourage active exploration, some layout-defining objects were partially obscured (e.g., table behind sofa) or deliberately placed along the same wall as the entrance, preventing full visibility from the initial entry point (see Fig. S2 for detailed layout of the rooms). Crucially, the rooms were designed so they could not be fully mapped from a single viewpoint, requiring participants to move around and explore different areas to build a complete cognitive map (Wolbers & Wiener, 2014).” (p. 4, paragraph 2).*

c) Show the distributions of path-RE and head-RE measures and the correlations between them.

We have included the residual correlation between path RE and head-direction RE in the

original manuscript, which demonstrates the correlation between path RE and head-direction RE after accounting for other predictors in the model: “*This bivariate approach was chosen to accommodate the correlation between these two exploratory behaviours after accounting for the predictors in the model (Experiment 1: posterior mean of residual correlation estimates = 0.18, 93%-highest posterior density interval (HPDI) = [0.10, 0.27]; Experiment 2: posterior mean of residual correlation estimates = 0.34, 93%-HPDI = [0.28, 0.39]).*” (p. 12, paragraph 3) To address the reviewer’s request, we have included additional analyses and visualisations of the distributions of path RE and head-direction RE, as well as the correlations between them. The updated results section now includes the following:

- Distributions of path RE and head-direction RE: “*Our virtual rooms successfully elicited varying levels of pre-room curiosity and post-room interest ratings (see Fig. S3 and Fig. S4 for the distribution of ratings within and across participants, respectively), as well as path RE and head-direction RE (see Fig. S5 for distribution of path RE and head-direction RE across participants).*” (p.11, last paragraph).

- Correlations between path RE and head-direction RE: “*Additionally, we observed a positive correlation between path RE and head-direction RE, both across and within participants, indicating that those who explored the space more extensively also tended to visually scan in more detail (see Supplementary Note 1 for further details on the relationship between path RE and head-direction RE).*” (p. 11, last paragraph – p. 12 paragraph 1).

d) Discuss how the distributions/correlations in c depend on time spent and on the type of room.

We further investigated how the correlations between path RE and head-direction RE depend on the time spent in each room and the room type. Our analysis revealed that the relationship between path RE and head-direction RE was negatively influenced by time spent inside the room in both experiments, which we included in the Supplementary Note 1: “*The models revealed that the relationship between path RE and head-direction RE was negatively impacted by the duration of time inside the room (Experiment 1: posterior mean of β = -0.0078, 93% HPDI = [-0.01, -0.0056]; Experiment 2: posterior mean of β = -0.0022, 93% HPDI = [-0.0037, -0.0008]). These results indicated that the longer the participant was inside the room, the weaker the relationship between the path RE and head-direction RE of their exploration. This could imply that when the duration is longer, participants may spend more time standing still and observing their surroundings or specific objects, rather than walking and observing simultaneously, leading to a decreased correlation between the two REs.*” (p. 5 in Supplementary Information file)

2. * It would be useful to have additional insights about the measures of pre-exploration curiosity and post-exploration interest. The authors should better describe: a) how were the ratings distributed within and across participants; b) how did the two ratings relate to each other (was more curiosity associated with more or less interest)?; c) related to b,

how do the authors think these two measures differ?; d) was “interest” a prospective measure(i.e., “how interested would you be to learn more about that room”?) or a retrospective one (“how interested were you while exploring the room?”).

Response: Thank you for highlighting the need for additional insights regarding the measures of pre-exploration curiosity (referred to as ‘pre-room curiosity’ in the manuscript) and post-exploration interest (referred to as ‘post-room interest’ in the manuscript). We address each point in detail below:

a) How were the ratings distributed within and across participants?

We have included plots showing the distributions of pre-room curiosity and post-room interest ratings within (Fig. S3) and across participants (Fig. S4) in the supplementary information file and refer to these distributions in the first paragraph of the Results (along with the distributions of the RE measures) (*p. 11, last paragraph*).

b) How did the two ratings relate to each other (was more curiosity associated with more or less interest)?

We analysed the relationship between pre-room curiosity and post-room interest ratings. Our findings indicate a positive correlation between the two measures, suggesting that higher curiosity before exploring a room was generally associated with higher interest after exploring the room. We have added the results to the supplementary information file (Supplementary Note 2) and reported in the Results section: “*Similarly, curiosity and interest ratings were positively correlated, suggesting that participants who reported higher levels of curiosity also found the rooms more interesting (see Supplementary Note 2 for details).*” (*p. 12, paragraph 1*)

c) Related to b, how do the authors think these two measures differ?

Pre-room curiosity and post-room interest capture different aspects of the participants’ engagement with the rooms. Pre-room curiosity reflects participants’ anticipation and motivation to explore the room before entering it, potentially based on prior information or expectations or driven by desire to address the uncertainty. In contrast, post-room interest captures participants’ *retrospective* evaluation of how engaging or intriguing they found the room after having explored it. While both measures are related, they tap into different temporal phases of the exploratory experience, with curiosity influencing the initiation of exploration behaviour and interest reflecting the outcome of that exploration. We have added this explanation to the main text: “*Here, pre-room curiosity and post-room interest capture different aspects of the participants’ engagement with the rooms. Pre-room curiosity reflects participants’ anticipation and motivation to explore the room before entering, driven by prior information, expectations, or a desire to resolve the uncertainty. In contrast, post-room interest reflects participants’ retrospective evaluation of how engaging or intriguing they found the room after exploring it. Interest ratings can be influenced by factors such as the novelty of objects, their arrangement, and how well they match participants’ expectations.*” (*p. 3, paragraph 1*)

d) Was “interest” a prospective measure (i.e., “how interested would you be to learn more about that room”?) or a retrospective one (“how interested were you while exploring the room?”).

“Interest” was a retrospective measure. In the task instructions, we asked participants to rate how interesting they found the room after exploring it. We have clarified this now in the manuscript (p. 3, paragraph 1; see above) and the Methods section: “*When deciding to leave, participants pressed “B”, leading to another rating display where they rated how interested they felt about the room on a Likert scale from 1 (“Not interesting at all”) to 10 (“Very much interesting”).*” (p. 5, paragraph 4).

- 3. It would be very helpful to have a bit more principled explanations about the associations reported. Some explanations are provided – e.g., about how animals may structure pathRE and headRE during visual exploration, and about the role of stress tolerance in moderating the relation between curiosity and path RE -- and I found them very helpful in understanding the data. But I missed a similar understanding of the distinct roles of pathRE and headRE, or of curiosity and interest in spatial memory (or “cognitive maps”). How do you think each measure contributes? What explains the negative association between memory and interest? And what do you think is the direction of the relationship: does higher curiosity/pathRE/headRE promote better spatial memory, or does a better capacity for spatial memory promote higher curiosity/pathRE/headRE? I realize that a lot of this discussion would need to be speculative, but it is nevertheless important to understand whether and how these measures are important.*

Response: Thank you for highlighting the need for more principled explanations about the associations reported in our study. We agree that the original manuscript did not fully unpack the distinct contributions of path RE, head-direction RE, curiosity and interest to cognitive map formation.

In response to this comment, we carefully went through the Discussion and throughout the Discussion we added relevant speculations about how our four measures (pathRE, head-direction RE, curiosity and interest) affect memory, and how the findings are in line with existing theories and empirical findings.

For example, regarding curiosity effects on memory, in paragraph 2 in Discussion (p. 21), we now added the following sentence: “*This positive influence of curiosity on spatial exploration and memory aligns with theories of curiosity as a cognitive enhancer (e.g., Gruber & Ranganath, 2019; Murayama, 2022; O’Keefe & Nadel, 1978) and extends previous research that has primarily focused on curiosity’s role in enhancing memory for verbal and semantic information (e.g., Fandakova & Gruber, 2021; Gruber et al., 2014; Kang et al., 2009; Murphy et al., 2021). Our findings bridge these theoretical perspectives with spatial cognition, demonstrating curiosity’s broader impact on cognitive processes.*” (p. 20, last paragraph)

We elaborated on the contrasting roles of curiosity and interest, speculating on how curiosity might drive more extensive, broad-based exploration (path RE), while interest may lead to more focused, selective attention (head-direction RE), which can also explain the negative association between memory and interest: *“Curiosity is often operationalised through exploration and information-seeking behaviours aimed at resolving uncertainty, which ultimately enhances learning and cognitive performance (e.g., Gottlieb & Oudeyer, 2018; Gruber & Ranganath, 2019; Murayama, 2022; Wang & Hayden, 2021). In contrast, post-room interest appears to engage more focused attention, driving visual exploration (head-direction RE) of specific features or objects. However, this interest-driven selective focus may occur at the expense of broader spatial encoding, which could potentially contribute to the observed negative trends in its association with path RE (Fig. 3) and cognitive map formation (Fig. 5B).”* (p. 22, paragraph 3) and *“One possibility is that curiosity drives movement through space (path RE) to discover novel parts of an environment, while subjective feelings of interest may promote visual exploration (head-direction RE) to examine these discovered parts more closely.”* (p. 23, paragraph 1)

Regarding the directionality of the relationship between curiosity/spatial exploration and spatial memory, based on our findings, these relationships are observed at the intra-individual level (see also our response to Reviewer 1, point 7), we believe it is more likely that heightened state curiosity and increased path RE promote better cognitive map formation, rather than reverse.

While we cannot entirely rule out the possibility of a bidirectional relationship, the intra-individual findings strongly support the notion that curiosity and exploration act as driving factors in enhancing spatial memory, rather than being a consequence of pre-existing memory capabilities. We have clarified this point in the revised manuscript to ensure the interpretation of our results is clear:

“The observed positive association between pre-room curiosity and cognitive map formation, particularly at the intra-individual level, suggests that heightened curiosity may facilitate more precise cognitive map formation.” (p. 21, paragraph 3)

and

“... we found that path RE – defined as the degree of spatial coverage in an open environment - was predictive of the precision of cognitive maps.” (p.22, paragraph 1).

Minor: Several Methodological details should be clarified:

4. *How did participants receive the 24-hour test in exp 1? Was this online or were they invited back in the lab?*

Response: We added this information now to the Supplementary Methods in the supplementary information file:

“In Experiment 1, participants were invited to return to the lab approximately 24 hours after completing the initial exploration task to finish the second part of the experiment. In this part of the experiment, participants took part in a surprise memory test.”
(Supplementary Information p. 3, paragraph 2).

5. *Exploration speed was limited to 2 eye-heights/second. What is an “eye height”?*

Response: In our study, “eye height” refers to the height of the eyes of the virtual agent in the virtual world, which corresponds to approximately 1.7 metres in the real world. Therefore, the exploration speed was limited to about 3.4 metres per second. We have clarified this in the Methods section:

“Movement speed in the virtual world was fixed at approximately 3.4 m/s, to simulate a brisk walking pace and ensure a standardised duration spent on the pathway.” (p. 5, paragraph 2).

6. ** No results were reported from the incidental memory test – are these analyzed separately, or did they not produce significant findings?*

Response: We decided not to include results of the incidental memory test in Experiment 1 in the current manuscript because the focus of the current manuscript is on the relationship between curiosity, spatial exploration, and cognitive map formation within the rooms themselves. The incidental memory test assessed memory for objects located in the pathway outside the rooms, which is not central to our primary research questions in this manuscript. Consequently, we chose to omit these results to maintain the focus of the manuscript and included the methodological details in the Supplementary Methods (see also our response to Reviewer 1, point 9). We have clarified this in the Methods section:

“Although other memory tests were conducted in both Experiment 1 and 2 (see Supplementary Methods for details of those tests), the findings from those tests will be reported elsewhere to maintain focus on the central research questions of this manuscript.” (p. 5, paragraph 3).

Reviewer 3

The purpose of this study is original in that it investigates the role of curiosity in spatial exploration and subsequent spatial memory. Indeed, on the one hand, the role of curiosity as a memory enhancer has already been shown (e.g., Gruber & Ranganath, 2019; Marvin & Shohamy, 2016; Murayama, 2022;), and on the other hand, curiosity states have been widely operationalized by exploration and information-seeking behaviors to resolve a lack of

information in a given activity/task, or a situation of uncertainty in a given environment, and ultimately be rewarded with learning that increases us cognitively (e.g., Gottlieb & Oudeyer, 2018; Murayama, 2022; Wang & Hayden, 2021).

Here, the major contribution is to demonstrate the link between these two literatures, using virtual environments to show that aroused curiosity predicts the amount of spatial exploration, and that curiosity and exploration are independently beneficial for spatial memory (learning a cognitive map of the environment). In this way, the initial intuitions of historical contributors such as Tolman (1948) or O'Keefe & Nadel (1978) to the study of spatial exploration and spatial memory in unfamiliar environments, i.e., curiosity, novelty seeking could be primary motives for exploration behaviors then requiring orientation and spatial memory skills.

Apart from this significant contribution, the other contribution is to offer a new VR-based experimental paradigm for exploring the links between curiosity and memory. Today, most studies only use verbal material (i.e., Trivia paradigm). Using this type of paradigm, it will be possible to study and even generalize the results obtained with verbal stimuli to non-verbal stimuli, such as visuospatial information. In addition, the strong positive relation between curiosity estimates and Roaming Entropy (RE) measures provides a promising way for probing curiosity states with objective measurements rather than subjective ones based on self-rating methods (that exhibit some limitations, e.g., Fastrich, G. M., Kerr, T., Castel, A. D., and Murayama, K. (2018). The role of interest in memory for trivia questions: an investigation with a large-scale database. *Motiv. Sci.* 4, 227–250. doi: 10.1037/mot0000087).

All in all, it's clear that this study is convincing and, in my expert opinion, deserves to be read and shared with readers of *Nature -Psychology communication*. However, I do have a few comments and points I'd like the authors to clarify, as follows.

Response: We thank the reviewer for their detailed comments and that they highly value our study. Below, we appropriately address all individual comments by the reviewer. We thank the reviewer for correctly summarising all key sections of our manuscript below and we note that, for ease of reading, we underlined the reviewer's points that they requested to clarify.

1) Regarding the state of the art:

1. p. 2. It is unclear why the authors contrast semantic knowledge acquisition and spatial exploration behaviors. Semantic knowledge has been shown to be extracted from curious spatial exploration. (Jirout, J., and Klahr, D. (2012). Children's scientific curiosity: In search of an operational definition of an elusive concept. *Dev. Rev.* 32, 125–160. doi: 10.1016/j.dr.2012.04.002 ; Jirout, J. J. (2020). Supporting early scientific thinking through curiosity. *Front. Psychol.* 11:1717. doi: 10.3389/fpsyg.2020.01717). For their study purpose, it's more the nature of the stimuli (verbal vs. non-verbal) that seems to matter, because in the end, curiosity-driven spatial exploration always aims for either perceptual and/or semantic stimulation. Moreover, there is a recent study with children

revealing the benefit from curiosity on spatial memory (Sivashankar, Y., Fernandes, M., Oudeyer, P. Y., & Sauzéon, H. (2024). The beneficial role of curiosity on route memory in children. Frontiers in Cognition, 3.)

Response: We appreciate the reviewer's suggestion to clarify our rationale for contrasting semantic knowledge acquisition and spatial exploration behaviours, including the references to relevant studies and recognise the importance of integrating these perspectives into our discussion.

As stated in the Introduction, our study aims to bridge two distinct literatures on curiosity: the animal literature, which focuses on spatial exploration behaviours as indicators of curiosity and their role in cognitive map formation, and the human literature, which primarily examined the role of curiosity in memory formation of semantic (non-spatial) information. While curiosity-driven spatial exploration can lead to the extraction of semantic knowledge (e.g., Ten et al., 2021, *Nature Communications*; Lydon-Staley et al., 2021, *Nature Human Behaviour*), the primary focus of our study is to understand how curiosity affects spatial exploration and cognitive map formation in humans.

To avoid confusion and better clarify our rationale, we have revised the relevant paragraph in the Introduction incorporating also the reviewer's proposed studies:

"This new research field has mostly focused on how curiosity drives the acquisition of semantic (non-spatial) knowledge (Eschmann et al., 2022; Gruber et al., 2014; Kang et al., 2009; Kobayashi et al., 2019; Lau et al., 2020; Lydon-Staley et al., 2021). This knowledge-driven type of curiosity is often referred to as epistemic curiosity (Berlyne, 1954). Additionally, some research has explored a more sensory-driven form of curiosity, known as perceptual curiosity, which can be sparked by novel, surprising or puzzling stimuli (Berlyne, 1954; Cohanpour et al., 2024; Collins et al., 2004; Jepma et al., 2012; Litman, 2000). Both types of curiosity can lead to exploratory behaviours that result in the extraction of semantic knowledge (e.g., Jirout, 2020; Lydon-Staley et al., 2021; Ten et al., 2021; see Jirout & Klahr, 2012 for review). However, despite these recent findings, the types of spatial exploratory behaviours observed across motile species and the impact of curiosity on spatial memory and formation of cognitive maps have received far less attention in human studies (Sivashankar et al., 2024)." (p. 2, paragraph 2)

- 2. p. 2 -3 : The authors refer extensively to their own and related work in the field of functional neuroscience, but work in the psychology of curiosity is worth mentioning, particularly to clarify the distinctions between perceptual vs. epistemic curiosity (e.g., Berlyne 1954, Litman, 2000, Collins, R. P., Litman, & Spielberger, 2004). The measurement of perceptual curiosity. Personality and individual differences, 36(5), 1127-1141). It would also be helpful if the authors could clearly define what is meant by curiosity, interest and novelty. At times, one gets the impression that curiosity and*

novelty are equated, which is confusing (cf. Murayama et. al., 2019). Sometimes, spatial and visual explorations are distinguished as if path-RE refers to spatial exploration while Head-RE refers to visual exploration. Definitions of distinctions and terminologies could prove useful in interpreting the double empirical dissociation reported in this study.

Response: Thank you for highlighting areas where further clarification is needed.

We agree that the early work in the psychology of curiosity distinguishing perceptual and epistemic curiosity should be mentioned and we have added this information accordingly in the Introduction (see our response to Point 1).

In the Introduction, we also now clearly define curiosity, interest, and novelty, and ensured throughout the manuscript that these constructs are dissociable and not equated:

For curiosity, we have provided a definition in the Introduction of the manuscript: *“the innate desire to seek out novel information”* (p. 2, paragraph 1). This definition emphasises curiosity as a motivational state that drives individuals to explore and acquire new information.

For interest, we have clarified the definition in the Introduction: *“participants also rated how interesting they felt about the room following the exploration phase (i.e., post-room interest).”* (p. 3, paragraph 1). Specifically, regarding the distinction between curiosity and interest, we have added *“Pre-room curiosity reflects participants’ anticipation and motivation to explore the room before entering, driven by prior information, expectations, or a desire to resolve the uncertainty. In contrast, post-room interest reflects participants’ retrospective evaluation of how engaging or intriguing they found the room after exploring it. Interest ratings can be influenced by factors such as the novelty of objects, their arrangement, and how well they match participants’ expectations.”* (p. 3, paragraph 1).

Novelty is often considered as a property of the stimuli and is usually thought of as a primer for eliciting curiosity that stimulates exploration (Berlyne 1950, 1954, 1966; O’Keefe & Nadel, 1978; Gruber & Ranganath, 2019). While novelty can trigger curiosity, they are not synonymous. For instance, people typically explore a novel environment more than a familiar environment, but the same novel environment can trigger varying levels of curiosity among different individuals, resulting in different amounts of exploratory behaviour. To clarify the distinction between curiosity and novelty, we added in the Introduction, where we describe the experimental design: *“These virtual rooms, being novel to all participants, were expected to stimulate varying degrees of curiosity and exploratory behaviours (Cen et al., 2021).”* (p. 2, last paragraph)

Distinction between and definitions of spatial exploration (path RE) and visual exploration (head-direction RE)

In our study, we measured two aspects of exploratory behaviours, spatial exploration and visual exploration. Here, spatial exploration refers to exploration of an environment

by walking or moving around it, and visual exploration refers to exploration of an environment by standing at a fixed spot and looking around it. We quantified spatial exploration using path RE which is computed on the participant's location data, and visual exploration using head-direction RE which is computed on the participant's head-direction data. We have clarified these definitions in the Introduction: *"Path RE measures spatial exploration by tracking participants' movements through the virtual rooms—specifically, how thoroughly they navigated different locations within each room. A higher path RE indicates a more extensive coverage, as participants moved through a larger portion of the room. In contrast, head-direction RE captures visual exploration by recording participants' head direction as they moved within the room. This measure reflects how broadly participants scanned the environment visually, turning their heads to observe their surroundings from different angles. Together, these two RE measures allow us to separately examine how spatial and visual exploratory behaviours contribute to the formation of cognitive maps."* (p. 3, paragraph 2).

Furthermore, we have acknowledged in the Limitations section in the Discussion that while path RE primarily measures spatial exploration and head-direction RE primarily measures visual exploration, there is a potential overlap in what each measure captures: *"Second, while we interpret path RE primarily as a measure of spatial exploration, capturing the participant's movement through space, it is important to acknowledge that it may also reflect some aspects of visual exploration, as participants encounter new visual stimuli while moving. Similarly, head-direction RE predominantly measures visual exploration, tracking the orientation of the participant's head as they scan their environment. However, it may also be influenced by the spatial layout of the environment. Despite these overlaps, most of the variance in path RE and head-direction RE is expected to correspond to spatial and visual exploration, respectively."* (p. 24, paragraph 2)

2) Regarding the study design:

3. Two studies are proposed using the same virtual application in which the participant freely explores a series of virtual rooms ($n=16$), and from which are measured:
 - the subjective state of curiosity before and after exposure to the virtual room
 - the participants' exploratory behaviour in each room is probed by Roaming Entropy (RE) measures (Freund et al., 2013) whose increase in value indicates a broad, or complex, exploration of the environment. Two RE measures are calculated: i) "Path RE" from the translational moves (via clicks of "W" and "S" letters on keyboard forwards/ backwards, respectively) for estimating the explored surface by the participant for each room (location data), ii) "head direction RE" from directional moves (via the tracking of the computer mouse changes in the direction of the movement and in the angle of the frame/field of view) for probing head direction moves

- *the memory performance of the path performed in each "room" is measured with a drawing task associated either with a yes/no recognition for the Exp. 1 (n =28) or with a free recall for the Exp. 2 (N=60). For the latter, the impact of trait curiosity (measured with the scale of Kashdan et al., 2020) is proposed. Why take a dimension out of the test and risk impoverishing its psychometric quality? If, as the authors presume, the excluded dimension is far removed from the construct of curiosity targeted in spatial exploration, it can serve as a measure of divergent validity. Also, for exp. 2., a new method of coding spatial memory performance is created to evaluate 4 dimensions in the formation of a cognitive map, i.e., Object Presence, Spatial Distortion and Rotation of Features, Relative Positioning, and Spatial Proportions. why only the structuring objects in the room and not also the other ones, that can have attractiveness value for a deeply exploration through curiosity?*
- *Overall, the changes in methods across the 2 experiments are rather confusing, and one comes to think that Exp. 1 is a pilot study and Exp. 2 the real study; so does Exp. 1 belong in the core of the paper, shouldn't it be in the supplementary material? This question is also motivated by the results that are reported as more reliable for Exp. 2 compared to those from Exp. 1, especially regarding the dissociation effect.*
- *The main hypotheses are: "i) the high states of pre-room curiosity positively affect the magnitude of spatial exploration while ii) the states of post-room curiosity are expected as positively related to both path-RE and head-direction RE.*

Response: The reviewer accurately summarised the key elements of our experimental design. Below we respond to the reviewer's proposed clarifications (underlined points).

The reviewer asks: "Why take a dimension out of the [curiosity trait] test [measured with the scale of Kashdan et al., 2020] and risk impoverishing its psychometric quality?" As the reviewer points out, we used the five-dimensional curiosity scale revised (5DCR) and excluded the dimensions of Overt and Covert Social Curiosity pertaining specifically to social aspects of curiosity because they are less relevant to the non-social, spatial exploration task used in our study. Our primary aim was to investigate the relationship between curiosity and spatial exploration behaviours. Therefore, we decided to exclude these social curiosity dimensions from our main analysis and to focus on the subscales (dimensions) of Joyous Exploration, Deprivation Sensitivity, Stress Tolerance, and Thrill Seeking that are more relevant to our experimental paradigm.

Moreover, while excluding Overt and Covert Social Curiosity might appear to risk impoverishing the psychometric quality of the 5DCR (Kashdan et al., 2020, *Personality and Individual Differences*), it is essential to recognise that the scale is designed to measure different facets of curiosity separately. In particular, Kashdan and colleagues

recommend computing the average item score for each dimension and analysing them individually.

Nonetheless, to explore the reviewer's concern about divergent validity, we conducted an additional analysis including all dimensions of the 5DCR scale to demonstrate the distinctiveness of the chosen sub-scales (see the figure below). Importantly, our main findings that Stress Tolerance mediated the curiosity-path RE relationship remained. As we only found a relatively small influence of social curiosity on the curiosity-driven spatial exploration in our paradigm we prefer to keep the original analysis in the main text as our paradigm did not include a social component. However, if the reviewer prefers, we are happy to add this analysis to the main text or supplemental data.

In addition, the reviewer asked about our rationale for focusing on layout-defining objects in the drawing memory test.

As the reviewer correctly pointed out, in Experiment 2, we developed a novel method to code spatial memory performance by evaluating four dimensions in the formation of a cognitive map: Object Presence, Spatial Distortion and Rotation of Features, Relative Positioning, and Spatial Proportions. Our primary goal in this drawing memory test was to investigate the accuracy of participants' cognitive maps of the rooms they explored.

We focused on key objects that define the layout of the room for scoring Object Presence because layout-defining objects serve as the foundational elements for all possible spatial relationships within the room. These objects provide critical reference

points that anchor the cognitive map and facilitate the organisation of other spatial information. As such, they are essential for assessing the overall accuracy of the cognitive map.

While we recognise that other objects in the room may have attractiveness value (e.g., paintings on the wall, books on the shelves and candles on the coffee table) and could influence exploration behaviour driven by curiosity, our primary interest was in evaluating the structural aspects of the cognitive map. The inclusion of all objects, regardless of their layout-defining properties, could introduce variability that might obscure the primary spatial relationship we aimed to measure:

“Object Presence: The dimension focused on the quantity and accuracy of key objects recalled from the virtual room. The focus was on major, layout-defining objects/landmarks (e.g., furniture, doors, windows) that contributed to the overall spatial configuration of the room. While the smaller details (e.g., plates on the table, books on the shelf and cakes in the display case) added natural details to the environment to encourage exploration, they were not considered essential for accurately sketching the room's overall layout.” (p. 9, paragraph 2)

Finally, the reviewer questioned whether both experiments should be included in the main manuscript and whether Experiment 1 should be moved to the supplemental results.

We prefer to keep both Experiment 1 and Experiment 2 in the main manuscript to emphasise the replication of findings (Nosek et al., 2022, *Psychological Science. Annual Review of Psychology*; Shrouf & Rodgers, 2018, *Annual Review of Psychology*) and the methodological evolution of our research. Experiment 1 was designed to explore the initial relationship between curiosity and spatial exploration. Based on the insights gained from Experiment 1, we expanded our investigation in Experiment 2 to include the formation of cognitive maps, thereby providing a more comprehensive understanding of how curiosity influences both exploration and cognitive map formation. This stepwise approach allowed us to refine our methods and build on our initial findings, thereby ensuring robustness and reliability in our results.

To clarify this, we revised the paragraph introducing the experiments: *“In Experiment 1, we examined the relationship between pre-room curiosity, post-room interest and exploratory behaviours. Building on these findings, Experiment 2 sought to replicate the results of Experiment 1 and expanded our investigation to explore how curiosity and exploration influence cognitive map formation. Critically, Experiment 2 included a memory test in which participants were required to draw a map of the layout of each room (see Fig. 1A, 5A). Importantly, map drawings allow more insight into the precision and content of memory representations (Bainbridge, 2022). Building on previous measures for sketch map drawings (Billinghurst & Weghorst, 1995; Blades, 1990;*

Newcombe, 1985; Rovine & Weisman, 1989; Tversky, 1981), our novel coding scheme to these drawings (see Fig. 5A and methods in the supplementary materials) allowed us to assess how curiosity states enhance cognitive map formation of novel rooms. Additionally, the larger sample size in Experiment 2 enabled us to explore the influence of individual differences in trait curiosity on curiosity-based exploration (Kashdan et al., 2020).” (p. 3, last paragraph)

3) Regarding the results:

- *The statistical treatments are well documented, and based on Bayesian models. The exploration duration is critically controlled in the correlations.*
- *As previously mentioned, the core of the main document may include only the more reliable and well-built results, i.e. results from the Exp. 2.*
- *A double dissociation between the effects of pre-room curiosity and post-room interest is observed on the two types of explorative behaviours : 1) there is a positive correlation between pre-room curiosity and path RE that is moderated by individual differences in trait curiosity (especially for individuals with high Stress Tolerance and high Deprivation Sensitivity; 2) There is a positive association between post-room interest and Head-direction RE.*
- *Cognitive map precision is positively related with pre-room curiosity and the two RE scores while it is negatively related with post-room interest.*
- *Pre-room curiosity has a positive impact on all the five [four] dimensions of cognitive map while Path RE positively influenced only object Presence of cognitive map formation.*

Response: We appreciate the reviewer’s thorough review to accurately summarise the results of our study. We’re grateful that the reviewer acknowledges the appropriateness of our statistical analyses.

Regarding the Results section, the reviewer only suggests (along with their comment above) to focus the main text on the more robust findings of Experiment 2. However, as outlined above, we believe it is important to present both experiments in the main text. Experiment 1 was designed to determine whether/how curiosity affects exploration, laying the groundwork for our study. Upon establishing this link, Experiment 2 then not only replicated these findings, reinforcing their reliability, but also extended the investigation into how curiosity and exploration influence cognitive map formation. Additionally, the larger sample size in Experiment 2 allowed us to explore how individual differences mediate the curiosity-exploration relationship. This series of experiments provides a comprehensive progression from foundational findings to more complex cognitive processes, thereby strengthening the reliability and depth of our conclusions. To clarify this, we added in the last paragraph of the Introduction (p. 3, last paragraph; see our response to Point 2 above).

4) Regarding the discussion/conclusion :

- [4.1] a double dissociation is shown, but at the end we wonder to which cognitive dimensions of curiosity this corresponds? Once again, neurofunctional models are taken up, but they don't shed any light on what could correspond to the priori curiosity - Path RE coupling, and the a posteriori curiosity - Head directions RE couplings. Psychological studies could help with interpretation but so could the path-integration model (e.g., Wolbers & Hegarty, 2010; Anastasiou, C., Baumann, O., & Yamamoto, N. (2023). Does path integration contribute to human navigation in large-scale space?. *Psychonomic Bulletin & Review*, 30(3), 822-842.)
- [4.2] In this study, room navigation is based on movement metaphors (keyboard for translational movement and mouse for directional movement), so movement is less embodied and mediated by repetitive "cognitive translations" of the metaphors. As a result, this may have reduced the sensory integration of the path, and hence of the cognitive maps constructed by the participants. Another consequence is that the use of metaphors may have reduced the sense of presence in virtual environments, and been deleterious to post-visit curiosity, thus disrupting cognitive map formation. In this respect, it would be interesting to have the differences between pre- and post-visit curiosity assessments, to see if the way of navigating the VR application had any consequences on subsequent curiosity states. Maybe, it is a possible explanation that the RE did not influence the 5 [4] dimensions of cognitive map?
- [4.3] The authors, in addition to highlighting their contributions, should also highlight the limits to be set in the future to support their (interesting) contribution.

Response: Thank you for your comment. We have carefully considered each point and are pleased to address them as follows:

4.1 Double dissociation and cognitive dimensions of curiosity

The observed dissociation may reflect distinct cognitive dimensions of curiosity and interest. Pre-room curiosity likely involves cognitive processes related to anticipation, novelty-seeking, and uncertainty resolution. These processes might motivate expansive spatial exploration to gather new information. This could explain its stronger association with path RE, which are consistent with theories of curiosity as a driver of exploration and information-seeking behaviours (Gruber & Ranganath, 2019, *Trends in Cognitive Sciences*; Murayama, 2022, *Psychological Review*). On the other hand, post-room interest may involve cognitive processes related to depth of processing, detail-oriented attention, and value assessment of encountered information. These processes might engage more focused attentional mechanisms, driving visual exploration to examine specific features or objects more closely, hence its stronger association with head-direction RE.

4.2 Impact of navigation metaphors on sensory integration and cognitive map formation

To address the concern about the potential reduction in the sense of presence and its impact on post-room interest, we analysed the differences between pre-room curiosity and post-room interest. The analysis showed that post-room interest ratings were not lower than pre-room curiosity ratings. The overall trend was that post-room interest ratings were generally higher than pre-room curiosity ratings. For Experiment 1, the posterior mean of the difference (post-room interest – pre-room curiosity) was 0.21 with a 93% HPDI of [-0.1, 0.52]; and for Experiment 2, the posterior mean of the difference was 0.32 with a 93% HPDI of [0.14, 0.5].

Our findings suggest that the use of keyboard and mouse controls for navigation did not significantly reduce the sense of presence or post-room interest. In fact, post-room interest ratings were generally higher than pre-room curiosity ratings, indicating that participants remained engaged and interested throughout the experiment. This observation counters the concern that movement metaphors negatively impacted post-room interest.

In addition, research has shown that desktop tasks using navigation metaphors similar to our study still result in significant spatial learning of the virtual environment (e.g., Hejtmanek et al., 2020, *Multisensory Research*). Moreover, navigation performance in virtual environments has been found to correlate strongly with real-world navigation performance, with studies showing significant correlations between virtual and real-world wayfinding performance. These findings collectively suggest that while our virtual environment may not provide the same level of sensory integration as real-world navigation, it still engages similar cognitive processes and provides a valid measure of spatial cognition and cognitive map formation (e.g., Coutrot et al., 2018, *Current Biology*; Coutrot et al., 2019, *PLOS ONE*).

Regarding the reviewer's comment on the influence of path RE on the dimensions of cognitive maps, our Bayesian analysis indicated that path RE did have some influence on the four dimensions of cognitive maps (Object Presence, Spatial Distortion and Rotation of Features, Relative Positioning, and Spatial Proportions). While the influence on Object Presence was the most reliable, the posterior distributions for the other three dimensions showed less credible effects, with part of the posterior distribution overlapping zero (see Supplementary Note 5 for more details). We also revised the relevant part in the Results to clarify this: "*Path RE positively influenced Object Presence and may have potential benefits on Spatial Distortion and Rotation of Features, reflecting a more nuanced contribution to cognitive map formation*" (p. 17, last paragraph).

Regarding the path integration model, we prefer refraining from adding further explanations based on this model for our current manuscript. The path integration model (Wolbers & Hegarty, 2010, *Frontiers in Human Neuroscience*; Anastasiou et al., 2023,

Psychonomic Bulletin & Review) focuses on spatial navigation through body-based cues like proprioception and vestibular feedback. However, in our study, participants navigated the virtual environment using a keyboard and mouse, limiting the involvement of body-based cues typically associated with path integration. Additionally, there were no pointing tasks or home location cues, which are crucial for studying path integration in spatial navigation.

4.3 Limitations of our study and future research directions

We agree with the reviewer that the Discussion would benefit from a section on “Limitations and Future Research Directions” and added the following section to the manuscript:

“Our study provides valuable insights into curiosity, exploration, and cognitive map formation, but several limitations warrant consideration. First, while our sketch map task is a validated measure of cognitive map accuracy, future studies could benefit from using complementary methods, such as navigation or wayfinding tasks, alongside the sketch map task. This would provide a more comprehensive assessment of cognitive map precision by capturing different aspects of spatial knowledge and its application. Second, while we interpret path RE primarily as a measure of spatial exploration, capturing the participant’s movement through space, it is important to acknowledge that it may also reflect some aspects of visual exploration, as participants encounter new visual stimuli while moving. Similarly, head-direction RE predominantly measures visual exploration, tracking the orientation of the participant’s head as they scan their environment. However, it may also be influenced by the spatial layout of the environment. Despite these overlaps, most of the variance in path RE and head-direction RE is expected to correspond to spatial and visual exploration, respectively. Lastly, a detailed analysis of environmental features and their relationship to interest and exploration presents an exciting avenue for future research, potentially elucidating the complex interplay between interest, visual exploration and environmental characteristics.” (p. 24, paragraph 2)

Reviewer #1 (Remarks to the Author):

The authors have done a thorough job of addressing my comments and those of the other reviews, and the manuscript is much improved as a result. I am happy to recommend publication.

Reviewer #2 (Remarks to the Author):

This revision comprehensively addresses my previous concerns. The added information about methodology, room layouts, the distributions of different measures, and the interpretation is detailed and helpful, and make this paper a strong and interesting addition to the literature. I have only a few remaining minor comments:

1. The participant samples are heavily biased toward females. In exp. 1, 25 of 28 participants are female and in exp. 2, 55 of 60 participants are! I apologize for not noticing this in the first pass, but the bias is quite extreme and should be acknowledged. I suggest that the authors include a note about it in the section on “Limitations”, and mention whether they have any reason to expect (or rule out) that gender/sex would affect the behaviors they study.
2. I appreciate the discussion of exploration duration, and I would like to ask for a bit more details on it. What did participants know at the start of the experiment about the rooms that were available to explore and the expected length of the experiment? E.g., were people expecting a 1 or 1.5 hour experiment, and were they told the total number of available rooms and/or the types of the rooms they would be able to access? Also, were participants required to explore all the rooms – and if not, did they explore all the rooms or did some people skip some rooms? If the latter, did people know the types of rooms they would have (and thus could skip some rooms strategically?) or were skipping decisions made on the spot (or by chance?). These details will help readers understand how people decided how much time to spend in each room.
3. The strong relation with Stress Tolerance is very intriguing, particularly since this personality measure shows up repeatedly in very different measures of curiosity and information gathering. I appreciate the citation of Rischall et al., which also finds Stress Tolerance to be an important predictor after exploring a wider range of personality traits. I wanted to alert the authors to a new paper that validates this result for a wider set of

tasks: <https://motivationsciencelab.com/publication/curiosity-in-cognitive-science-and-personality-psychology-individual-differences-in-information-demand-have-a-low-dimensional-structure-that-is-predicted-by-personality-traits/> The paper is in press at PNAS; there is no obligation to cite it, but I wanted to mention it in case the authors find it interesting to consider and/or comment on the relationship with their paper.

Thank you again for a comprehensive revision and congratulations on an excellent paper!

Jackie Gottlieb

Response: We thank the reviewer for her positive comments on our revised manuscript. We address the comments point-by-point below:

1. We agree with the reviewer that our samples were heavily skewed toward female participants (i.e., 89% and 92% women in Experiments 1 and 2, respectively). This arose from limitations in the available participant pool during our data collection.

To address this point, we have added the following text to the Limitations section (page 24, paragraph 1): *“Third, both our experiments were heavily skewed toward women and future studies on curiosity-based exploration should aim for an equal gender balance to investigate whether the observed findings are generalisable across gender.”*

2. In the task instructions given before the experiment, participants were informed they would be exploring 16 different rooms, with no specific time limit imposed. During the familiarisation phase, they explored two example rooms to become acquainted with the procedure. We have now added this information in the Methods (page 4, last paragraph): *“At the start of the experiment, participants were informed they would be exploring 16 different rooms. During the familiarisation phase, they explored two example rooms to become acquainted with the procedure. This allowed them to get a sense of the types of rooms and environments they would encounter (e.g. rooms representing real-world settings), as well as the overall scope of the exploration task.”*

Participants were required to visit all 16 rooms, with the room order randomized across participants. To clarify these room visitation requirements, we have added the following to the Methods section (page 5, paragraph 2): *“The exploration*

phase closely followed the familiarisation phase, where participants had to explore all 16 rooms in a randomised order with no specific time limit imposed for each room.”

Participants could spend as much or as little time as they wished in each room. Overall, they rarely skipped rooms (i.e., spending minimal time exploring them entirely), except for one participant in Experiment 1. This participant was excluded from analysis due to incomplete exploration data, as they spent very little time in 12 of the 16 rooms (e.g. only briefly opening the door without fully entering or exploring the room contents). We have revised the sentence indicating this exclusion in the Participants section of the Methods for clarity (page 3, last paragraph): *“Furthermore, one participant was removed from analysis in Experiment 1 due to incomplete exploration data, having spent very little time in 12 of the 16 rooms (e.g. only briefly opening the door without fully entering or exploring the room contents).”*

3. Thank you for alerting us to the fascinating new work by Jach and colleagues. We have added this work to the Discussion (page 23, paragraph 2): *“Our finding that individuals with higher Stress Tolerance are more likely to act upon their curiosity and therefore explore novel, uncertain environments is in line with recent findings that also showed that individual differences in Stress Tolerance were associated with more effective information sampling strategies (Rischall et al., 2023) and more engagement in uncertainty-driven exploration (Jach et al., 2024).”*

Reviewer #3 (Remarks to the Author):

It's a version that satisfies my comments and the authors have modified or justified their answers. It's a great contribution to the field of curiosity!